# Increasing summer net $CO_2$ uptake in high northern ecosystems inferred from atmospheric inversions and comparisons to remote sensing NDVI

Lisa R. Welp[1,*], Prabir K. Patra[2], Christian Rödenbeck[3], Rama Nemani[4], Jian Bi[1], Stephen C. Piper[1], and Ralph F. Keeling[1]

[1]Scripps Institution of Oceanography, University of California San Diego, La Jolla, California, USA

[2]Agency for Marine-Earth Science and Technology, Yokahama, Japan

[3]Max Planck Institute for Biogeochemistry, Jena, Germany

[4]NASA Ames Research Center, Moffet Field, California, USA

*now at: Purdue University, West Lafayette, IN, USA

*Correspondence to:* L. R. Welp (lwelp@purdue.edu)

**Abstract.** Warmer temperatures and elevated atmospheric $CO_2$ concentrations over the last several decades have been credited with increasing vegetation activity and photosynthetic uptake of $CO_2$ from the atmosphere in the high northern latitude ecosystems: the boreal forest and Arctic tundra. At the same time, soils in the region have been warming, permafrost is melting, fire frequency and severity are increasing, and some regions of the boreal forest are showing signs of stress due to drought or insect disturbance. The recent trends in net carbon balance of these ecosystems, across heterogeneous disturbance patterns, and the future implications of these changes are unclear. Here, we examine $CO_2$ fluxes from northern boreal and tundra regions from 1985 to 2012, estimated from two inverse models (RIGC and Jena). Both used measured atmospheric $CO_2$ concentrations and wind-fields from interannually variable climate reanalysis. In the Arctic zone, the latitude region above 60°N excluding Europe (10°W – 63°E), neither model finds a significant long-term trend in annual $CO_2$ balance. The boreal zone, the latitude region from approximately 50°N to 60°N, again excluding Europe, showed a trend of 8–11 Tg C $yr^{-2}$ over the common period of validity from 1986 to 2006, resulting in an annual $CO_2$ sink in 2006 that was 170–230 Tg C $yr^{-1}$ larger than in 1986. This trend appears to continue through 2012 in the Jena inversion as well. In both latitudinal zones, the seasonal amplitude of monthly $CO_2$ fluxes increased due to increased uptake in summer, and in the Arctic zone, also due to increased fall $CO_2$ release. These findings suggest that the boreal zone has been maintaining and likely increasing $CO_2$ sink strength over this period, despite browning trends in some regions, changes in fire frequency and land use. Meanwhile, the Arctic zone shows that increased summer $CO_2$ uptake, consistent with strong greening trends, is offset by increased fall $CO_2$ release, resulting in a net neutral trend in annual fluxes. The inversion fluxes from the Arctic and boreal zones covering the permafrost regions showed no indication of a large-scale positive climate-carbon feedback caused by warming temperatures on high northern latitude terrestrial $CO_2$ fluxes from 1985 to 2012.

## 1 Introduction

The high northern latitudes, including the tundra and boreal forest regions, are particularly vulnerable to the effects of climate change as this region has been experiencing dramatic changes in recent climate. Warming in northern ecosystems results in many physical and ecological changes that have consequences for carbon cycling (Chapin, 2005; Hinzman et al., 2005; McGuire et al., 2009; Serreze et al., 2000; Smith and Dukes, 2012; Walther, 2010; Wu et al., 2012). Annual mean surface air temperatures over land increased by 0.64°C per decade north of 60°N from 1979 to 2008, roughly twice the rate of 0.33°C per decade for the northern hemisphere as a whole (ACIA, 2004; Bekryaev et al., 2010; Wolkovich et al., 2012). This northern polar amplification has been attributed to ice/snow-albedo feedbacks (Cess et al., 1991; Qu and Hall, 2007; Serreze and Barry, 2011). Minimum sea ice extent in the Arctic Ocean has declined rapidly (Comiso et al., 2008) with feedbacks and teleconnections on the continental areas as well (2016; Francis et al., 2009). Impacts in the northern regions are predicted to intensify, as climate scenario modeling projects further arctic temperature increases of 5–7 °C by the end of this century (ACIA, 2004; Ewers et al., 2005) and atmospheric $CO_2$ concentrations continue to rise at a rate approaching 2.0 ppm per year.

Tundra ecosystems and boreal forests hold large stores of carbon in soil organic matter buried in cold or frozen permafrost soils. It is estimated that 1,400 to 1,850 Pg C are stored in high northern latitude soils and another 60 to 70 Pg C in above and below-ground vegetation (Gedney et al., 2006; Keenan et al., 2013; McGuire et al., 2009). The natural turnover time of this carbon is very slow, but there is a risk that warmer temperatures will increase microbial respiration rates and expose previously frozen organic matter to decomposition by melting the permafrost (Cao et al., 2010; Johnstone et al., 2010; Schuur et al., 2008; Trahan and Schubert, 2015). Over the past several decades there has been a measureable trend to earlier spring snowmelt and surface soil thaw (Brown et al., 2013; McDonald et al., 2004; Smith, 2004) and increases in permafrost borehole temperatures (Romanovsky et al., 2010) demonstrating changes in the thermal stability of northern circumpolar soils. It has been speculated that warming could trigger a massive release of carbon from these soils, in the form of $CO_2$ and $CH_4$, leading to a positive climate-carbon feedback (Pastick et al., 2015; Schuur et al., 2008). This has been nicknamed the Arctic 'carbon bomb' in the popular media (Treat and Frolking, 2013).

However, warming and the associated lengthening of the growing season encourages plant growth in these otherwise temperature-limited areas, as does increased atmospheric $CO_2$ fertilization (Lloyd and Farquhar, 1996; Wickland et al., 2006; Yi et al., 2009) and increased nitrogen deposition (Baird et al., 2012; Barber et al., 2000; Bunn and Goetz, 2006; Goetz et al., 2005; Holland et al., 1997; Soja et al., 2007). The net carbon balance of increased plant growth and increased soil respiration is unclear, but has important consequences for predicting carbon-climate feedbacks (Abbott et al., 2016; Koven et al., 2015).

Measurements of atmospheric $CO_2$ concentrations at Barrow, Alaska by Keeling et al. (1996) provided evidence for increased photosynthetic activity and net primary production (NPP) at northern latitudes from 1960 through 1994. The changes were attributed to increased $CO_2$ uptake by vegetation during spring and summer, leading to earlier drawdown and larger seasonal amplitudes of atmospheric $CO_2$ concentrations (Barichivich et al., 2014; Keeling et al., 1996; Randerson et al., 1999). This perspective is also supported by satellite observations of an increase in vegetation greenness at northern latitudes (Myneni et al., 1997) and global ecosystem process models suggesting that northern ecosystems have become more productive as a result of combined changes in temperature, $CO_2$ concentration and nitrogen availability (Kimball et al., 2007; McGuire et al., 2001; Wang et al., 2012). An updated perspective on the northern $CO_2$ cycles from Barrow data and from repeated airborne surveys of the mid troposphere showed 50% increase in the amplitude from 1960 to 2010, implying a significant increase in northern ecosystem growing season $CO_2$ uptake over the last several decades (Graven et al., 2013; Ueyama et al., 2014).

Since the late 1990s, however, some indicators of ecosystem function suggest that the terrestrial biosphere response to recent climate change in the high northern latitudes may be different from the previous few decades, and that terrestrial $CO_2$ uptake has since slowed down or even turned to a net source. Analysis of the changes in the seasonality of atmospheric $CO_2$ suggests that temperature-induced late summer drought may be increasing fall $CO_2$ release and offsetting enhanced spring $CO_2$ uptake (Angert et al., 2005; Piao et al., 2008; Rozendaal et al., 2009). Piao et al. (2008) estimated that current warming during autumn increases respiration in northern ecosystems enough to cancel 90% of the increased spring $CO_2$ uptake. While these studies provide important insights into changing ecosystem function, changes in $CO_2$ seasonal cycles in the atmosphere depend not just on surface fluxes but also variations in atmospheric circulation (Higuchi, 2002).

Vegetation productivity and distribution have also changed during this same period. The treeline has advanced northward and woody shrub colonies have expanded in the tundra zone displacing less productive species (Goetz et al., 2005; Lloyd et al., 2005; Pearson et al., 2013; Stoll and Ortega, 2013; Sturm, 2005; Tape et al., 2006). Within the boreal zone, satellite observations show large areas of the boreal forest not disturbed by fire have been 'browning' since 2000 as observed by Normalized Difference Vegetation Index (NDVI) measurements (Goetz et al., 2005; 2007; Verbyla, 2008; 2011; Zhang et al., 2008). These studies are mostly based on either trends in maximum or growing season integrated NDVI or NDVI-derived LAI. One statistical analysis suggests the browning trends in the Alaskan boreal forest have been ongoing for the last three decades (Forkel et al., 2013). These results have been consistent with ground observations which also report widespread tree mortality caused by insect outbreaks due to warmer winter temperatures (Kurz et al., 2008) and drought (Hogg et al., 2008; Peng et al., 2011). However, an updated NDVI processing algorithm in the NDVI3g product shows overall more areas greening than the older version, with the largest greening in western Eurasia (Xu et al., 2013; Zhu et al., 2016) but with browning still occurring in parts of North America (Xu et al., 2013). Park et al. (2016) estimates that the overall greening, or increase in

growing season integrated NDVI, is equivalent to a 20.9% gain in productivity since 1982 and the smaller areas of browning are equivalent to a 1.2% loss of productivity. Furthermore, these trends in greening and productivity seem to be independent of shifts in the start and end of the growing season and growing season length (Park et al., 2016).

It is important to determine the net carbon balance of these large northern regions to see if they have been increasing or decreasing in $CO_2$ sink strength, or perhaps transitioning to a net $CO_2$ source. Approaches used to estimate the net $CO_2$ fluxes of large areas include forest inventories, atmospheric inversions, and process-based models. Each of these methods has its strengths and weaknesses. Atmospheric inversions can infer the global, continental and sometimes regional-scale fluxes of $CO_2$ between the atmosphere and

the land biosphere and the oceans, by analyzing the temporal and spatial records of atmospheric $CO_2$ change (Enting, 2002). Inversions have the advantage of including the effects of disturbance and changing vegetation patterns, but are limited to the period of sufficient $CO_2$ concentration observations and are best suited to resolving continental-scale fluxes. Few inversion analyses have specifically focused on the high northern latitude terrestrial ecosystems. Zhang et al. (2013) aggregated the inversion fluxes from five

different models into Eastern Canada and Western Canada plus Alaska. The inversions examined showed consistent increases $CO_2$ uptake in Eastern Canada and no long-term trend in Western Canada plus Alaska.

Atmospheric inversion $CO_2$ fluxes can provide useful validation metric for other methods of ecosystem monitoring because they resolve the net effect of above and below ground carbon fluxes. Repeat forest inventories are useful for identifying trends in forest productivity over time. However, they are limited to

detecting trends in above-ground carbon only, not changes in below-ground carbon. Forest inventory studies have found drought-induced tree mortality and above ground carbon loss in Canada, with the western region most affected (Hogg et al., 2008; Ma et al., 2012; Michaelian et al., 2010; Peng et al., 2011).

Process-based modeling studies attempt to account for above and below-ground carbon changes while providing full spatial coverage, and are therefore capable of simulating net ecosystem fluxes that can be

compared to atmospheric inversions. Several process-based modeling studies have concluded that the Arctic tundra and boreal forests have been decreasing sinks or increasing sources since the 1980s due to climate effects, namely warmer temperatures increasing soil organic matter decomposition, and increased fire and insect disturbance, offsetting increased $CO_2$ uptake driven by $CO_2$ fertilization (Bradshaw and Warkentin, 2015b; Hayes et al., 2011; McGuire et al., 2010)., though McGuire et al., (2012) found that 6

our of 11 processes based models (global and regional) predicted a strengthening carbon sink (including CH4 contributions) in the 2000s compared to the 1990s. Processes including permafrost melt, hydrologic changes, nutrient dynamics, and fire emissions are critical to predicting any changes in the net $CO_2$ fluxes from the northern regions, but difficult to include in the models with much certainty (Abbott et al., 2016; Harden et al., 2012). Abbott et al. (2016) conducted a survey of experts in the field and found that the

overwhelming opinion was that any increases in biomass are not going to be enough to offset carbon

releases from permafrost melt by 2040 nor 2100. These models and expert predictions can be directly challenged with results from atmospheric inversions.

In this study, we examine large regional-scale temporal variability in terrestrial $CO_2$ fluxes from two atmospheric inversions using interannually variable atmospheric transport from 1985 to 2012. We focus on trends in the carbon uptake of the land biosphere north of approximately 50°N.. The primary objective of this study is to evaluate temporal changes in the annual and seasonal land biosphere $CO_2$ fluxes. We determine in what months surface $CO_2$ fluxes have likely changed, i.e. increased summer uptake or winter release. We further examine NDVI, air temperature trends, and correlations with $CO_2$ fluxes to provide some spatial and process context for temporal changes in the inversion fluxes.

## 2 Methods and data analysis
### 2.1 Inversion models

We compared two different atmospheric inversions: the RIGC inversion and the Jena $CO_2$ inversion (s85v3.6). The RIGC inversion method was adapted from Rayner et al. (1999), and largely followed the TransCom-3 protocol (Gurney et al., 2003). The RIGC model uses a 64-region time-dependent inverse method to infer carbon source/sink estimates based on the method of Patra et al. (2005). The RIGC inverse calculation starts with *a priori* fossil-fuel emissions and terrestrial and oceanic fluxes which are then optimized to match observations. For the *a priori* fluxes, terrestrial exchanges were taken from the CASA monthly output (Randerson et al., 1997) and monthly-mean oceanic exchanges from Takahashi et al. (Takahashi et al., 2002) as in TransCom-3 protocol (Gurney et al., 2003). Total anthropogenic $CO_2$ emissions were derived from the Oak Ridge National Lab monthly fossil fuel estimates from CDIAC (Boden et al., 2009) plus bunker fuel and non-fuel oxidation estimates from the Emissions Database for Global Atmospheric Research (EDGAR) (Oliver and Berdowski, 2001). RIGC uses the NIES/FRCGC (National Institute for Environmental Studies/Frontier Research Center for Global Change) global forward transport model driven by interannually varying (IAV) meteorology from the NCEP reanalysis and the GLOBALVIEW-$CO_2$ product to derive residual *a posteriori* land and ocean surface fluxes for the 64 inversion regions.

In the RIGC inversion, the GLOBALVIEW-$CO_2$ input was limited to the 26 stations, which have nearly continuous $CO_2$ observation from 1985 to 2006 (Table 1). Among these are stations that document changes in high northern latitudes, including Barrow (71°N), Alert (82°N), Station M (66°N), Cold Bay Alaska (55°N), Shemya (52°N), and Cimone (44°N). A selected set of stations was used to avoid creating spurious trends in the inversion results from adding new stations mid-way through the inversion period. All selected stations had at least 71% of months sampled at least once and came online by 1989. Stations north of 39°N had 84% to 100% monthly coverage. The resulting fluxes from the RIGC inversion are valid from 1986 to 2006, after removing years at the beginning and end for 'edge effects' from the inversion setup.

Compared to the RIGC model, the Jena inversion, version s85_v3.6 (Rodenbeck, 2005), uses a slightly different set of 19 stations selected to completely cover the 1985–2012 estimation period, but includes all of the same stations north of 50°N (Table 1). It uses individual measurements from various sampling networks, without smoothing or gap filling. Fluxes are estimated at the grid-scale resolution (approximately 4° latitude by 5° longitude), to reduce aggregation errors. However, to counteract that the estimation would be underdetermined, spatial and temporal a-priori correlations are imposed, smoothing the estimated flux field on scales smaller than about 1 week and about 1600 km (land, in longitude direction), 800 km (land, latitude), 1900 km (ocean, longitude), or 950 km (ocean, latitude), respectively. Land flux adjustments are spatially weighted with a productivity proxy, the long-term mean NPP from the LPJ terrestrial biospheric model (Sitch et al., 2003). Prior fluxes comprise anthropogenic $CO_2$ emissions from EDGAR v4.2 (EDGAR, 2011), a constant spatial flux pattern on land (time-mean NEE from the LPJ model), and an ocean-interior inversion by Mikaloff Fletcher et al. (2006), with a mean seasonal cycle of ocean fluxes from Takahashi et al. (2002). The Jena inversion uses the TM3 global atmospheric transport model driven by interannually varying meteorology from the NCEP reanalysis. The 4 x 5 degree gridded *a posteriori* land and ocean surface fluxes are aggregated to our larger analysis regions. Resulting fluxes are valid from 1985 to 2012.

## 2.2 Datasets

We compared $CO_2$ fluxes with satellite-based normalized difference vegetation index (NDVI) data over the same time period and with land temperature records. NDVI is a proxy for photosynthetically active above-ground biomass, calculated from the visible and near-infrared light reflected by vegetation. It has dimensionless units and varies from a value of 0 for no vegetation to a value of 1 for the highest density of green leaves. We used NDVI data produced by NASA's Global Inventory Modeling and Mapping Studies (GIMMS version 3g) from measurements of the Advanced Very High Resolution Radiometer satellite and supplied at the monthly, 1 x 1 degree resolution (Pinzon and Tucker, 2014). Winter NDVI data was excluded from this analysis because of the confounding influence of snow (Myneni et al., 1997). We defined growing season NDVI (Zhou et al., 2001) as the sum of monthly NDVI from April to October following the example of earlier work. Comparisons with recent satellite measurements of solar induced fluorescence show that the seasonality of NDVI may not capture the seasonality in GPP (Walther et al., 2015), but we focus on interannual variability of growing season sums and maximum July values in this study.

We used monthly mean temperature anomalies from the NASA GISS 2 x 2 degree gridded dataset to compare to $CO_2$ fluxes and NDVI variability (Hansen et al., 1999). Temperature anomalies are computed by subtracting the 1951 to 1980 mean. Throughout this manuscript, we abbreviate seasonal means by 'MAM' (March, April, May), `JJA' (June, July, August), `SON' (September, October, November), and `DJF' (December, January, February).

We examined estimates of fire $CO_2$ emissions from 1985 to 2000 from the RETRO compilation and from 1997 to 2012 from the GFEDv4 model (Giglio et al., 2013; Schultz et al., 2008).

### 2.3 Analysis approach

Our focus is primarily on the interannual variability in the $CO_2$ fluxes which is considered more robust across inversions than the absolute values of the mean fluxes (Baker et al., 2006). For that reason, we look at anomalies from the long-term mean values of each inversion model.

We focused our analysis on land carbon fluxes in two roughly zonal bands at high northern latitudes partly based on the TransCom regional boundaries defined by Gurney et al. (2003). Figure 1 shows the regions of Boreal Asia (BA) and Boreal North America (BNA) that we aggregated into what we refer to as the 'boreal zone' roughly between 50°N and 60°N and the 'Arctic zone' north of 60°N. Note here that while we refer to the boreal zone as roughly '50°N to 60°N', the southern boundary is not defined at the 50°N latitude, but follows the irregular southern boundary of boreal forest (stippled area in Fig. 1), nor does it include the entire boreal forest as the northern boundary extends well into the 'Arctic zone' north of 60°N. We decided to omit the European (EU) land region from our zonal analysis for two reasons. First, the TransCom protocol followed by the RIGC inversion does not separate northern Europe at 60°N like it does for BA and BNA, rather northern EU section is everything north of 50°N. Second, the EU region includes a relatively small fraction of the tundra and boreal forest ecosystems compared to BA and BNA, and the forest area is highly managed. Our focus is how the less intensively managed ecosystems of the north have been responding to climate change and examining the boreal zone and the arctic zone should maximize any potential signals of change. A similar approach of excluding EU was used in the Arctic analysis of McGuire et al. (2009).

The atmospheric inversion approach taken in this study is unlikely to reliably separate influences from different longitudinal regions within the latitude bands discussed here. Our focus on the longest records possible, from sparse atmospheric $CO_2$ observations starting in the 1980s, compromises the spatial resolution of the inversion fluxes. Rapid atmospheric mixing of a few weeks around latitude bands makes it hard to separate fluxes for example from North America and Eurasia. For that reason, we check the EU and Northern Ocean (NO) regions for any trends that might be offsetting trends in what we define as the 'Boreal zone' and the 'Arctic zone' caused by spatial errors in the assignment of surface fluxes by the inversion analyses that could complicate our interpretations of the data. We performed the trend analysis for two periods: from 1986-2006 when we have inversion results for both models, and from 1985–2012 for just the Jena s85 inversion.

We examine trends in the monthly, seasonal, and annual net ecosystem exchange (NEE) fluxes from the inversion models. Amplitudes of the annual seasonal cycle in $CO_2$ fluxes were calculated from the maximum and minimum monthly mean fluxes within each calendar year as: flux amplitude = maximum

NEE - minimum NEE.  The flux amplitude is indirectly related to the amplitude in atmospheric $CO_2$ concentrations, as the atmospheric concentration is roughly the integral of the monthly fluxes. It is unnecessary to detrend the time series of fluxes from the models prior to calculating the flux amplitude, unlike the concentration amplitude which has a persistent long-term trend from anthropogenic $CO_2$ emissions.  We also examine the latitudinal gradient of the trends in the seasonal fluxes in $\sim4°$ latitude bands from the gridded Jena inversion.  This analysis was not possible with the RIGC inversion because of the larger basis regions.  This approach attempted to answer the question of whether summer uptake is increasing or fall respiration (or both) and how that might change with latitude.

Two methods were used to calculate the slopes of long-term trends and statistical significance of trends, linear least squares and Mann-Kendall tests.  Trends were considered significant if they passed the 90% confidence level (p-values < 0.1).  The Model I linear region analysis (LSQ) was done in Matlab using the function 'lsqfity' developed by Peltzer (2000) based on Bevington and Robinson (1992).  The non-parametric monotonic Mann-Kendall trend test (M-K) with Sen's slope was also done in Matlab using the function 'ktaub' developed by Burkey (2006).   Results of both tests are presented in Table 2.   The independent tests generally agree on slope and significance of trends.

### 3  Results
### 3.1  $CO_2$ flux trends
### 3.1.1 Arctic zone (>60°N)

The arctic zone containing the tundra region showed no significant trend in annual $CO_2$ uptake (Fig 2a, Table 2) from 1986–2006 in either inversion.  The longer period, 1985–2012, in the Jena inversion did show a small but significant trend toward increased uptake of an extra 4 Tg C $yr^{-1}$.  Anomalously strong annual $CO_2$ uptake occurred in years 1990 and 2004 in the RIGC inversion and strong $CO_2$ release in 1996. These large anomalous fluxes were not present in the Jena inversion.

The Jena and RIGC inversions differ in their mean seasonal cycle in the arctic zone, with the RIGC inversion yielding peak $CO_2$ uptake approximately twice that of the Jena inversion (Fig 3b). The seasonal amplitude and phase has previously been found to differ among inversion models (McGuire et al., 2012). These differences are not unexpected given the differences in atmospheric transport (including vertical mixing and leakage across latitudes), a priori fluxes, observational network inputs, and model structure between the inversion models.  In this analysis we try to focus on the most robust features were the models do tend to agree on the interannual trends in anomalies from the mean.  Trends in monthly net $CO_2$ flux, computed with the method of Randerson et al., (1997), reveal increasing uptake in July in both inversions and stronger releases in September, October, and November (Fig. 3d). These seasonal changes largely cancel in the annual net fluxes, but contribute to increasing $CO_2$ flux amplitudes, computed as the difference between the maximum and minimum monthly $CO_2$ fluxes, by $\sim1.0\%$ $year^{-1}$ relative to the mean seasonal amplitude from 1986 to 2006 for both inversions (Fig 4, Table 2).  Figure 5 shows the annual

values of the July $CO_2$ flux in Pg C $yr^{-1}$ over this record. This is directly related to July trend data in Figure 3d. On a per area basis, this translates to an increase in July peak summer $CO_2$ uptake of 0.007–0.013 gC $m^{-2}$ $day^{-1}$ $yr^{-1}$, depending on the inversion used, averaged over the entire zone or a ~10% increase in peak summer $CO_2$ uptake over these 21 years. The trends over 1985–2012 are similar at 0.007 gC $m^{-2}$ $day^{-1}$ $yr^{-1}$ for the Jena inversion (Table 2).

### 3.1.2 Boreal zone (50°N – 60°N)

The boreal zone shows a trend towards increasing annual net $CO_2$ uptake in both inversions (Fig 2b, Table 2). From 1986 to 2006, the trend in the RIGC inversion was 10 Tg C $yr^{-1}$ with a p-value <0.1. The Jena inversion resulted in a similar trend of 8 Tg C $yr^{-1}$, but did not meet the criteria for significance, p>0.1 (Table 2). The most noticeable difference between the inversions is that the RIGC inversion predicted an anomalous release of $CO_2$ in 1994 that was not confirmed by the Jena inversion. Over the longer period from 1985–2012, the Jena inversion predicts the same trend toward greater $CO_2$ uptake with a slope of 7–8 Tg C $yr^{-1}$ and a p-value <0.1 (Fig. 2b, Table 3).

The Jena and RIGC inversions resulted in similar mean seasonal cycles of the monthly net $CO_2$ fluxes, but the seasonal amplitude in the Jena inversion was slightly larger (Fig. 3a). The trends in the monthly fluxes show increasing $CO_2$ uptake in the growing season, and in the case of the Jena inversion, increasing $CO_2$ uptake in the spring and release in the fall (Fig. 3c). There was a corresponding increase in the seasonal amplitude of net $CO_2$ flux of 0.4% $yr^{-1}$ (p=0.04) estimated by the Jena inversion, but not in the RIGC inversion (Fig. 4b and Table 2). Figure 5 shows the time series $CO_2$ flux in July (month of peak flux) over this period. Both models show an increase in July $CO_2$ uptake although they don't agree on anomalies from year to year. In Figure 3cd, both models also show an increase in the fall $CO_2$ release in the northern land regions, but the Jena inversion attributes this mostly to the boreal zone, whereas the RIGC inversion attributes it mostly to the arctic zone.

### 3.2  Europe and Northern Ocean fluxes and fossil-fuel emissions

For completeness, we also show the time series of $CO_2$ flux trends, both net annual and seasonal amplitude from the 55°N to 80°N region of Europe (EU) and the northern ocean (NO) to be sure that fluxes in these regions are not compensating for fluxes in our analysis of the BA+BNA regions (Fig. 6). There were no offsetting positive trends in the annual net flux of $CO_2$ or negative trends in the seasonal amplitude in EU from 1986–2006 and none were statistically significant (Table 2). Likewise the NO flux trends are insignificant with the exception of the seasonal amplitude trend in the RIGC inversion of -2.4% $yr^{-1}$. This was statistically significant using the modified Mann-Kendall p-test, but the mean amplitude of the ocean flux (~0.1 Pg C $yr^{-1}$) is much too small to offset gains in the land flux amplitude in the BA+BNA regions (mean 4.6–9.9 Pg C $yr^{-1}$ for north of 60°N and mean 11.4–13.9 Pg C $yr^{-1}$ for 50°N–60°N). There is no

indication that inversion-resolved trends in the EU and NO regions in the north of 50°N zone are forcing offsetting trends in the BA+BNA regions, however, we cannot rule out misallocation of fluxes among the inversion regions used in this study.

The RIGC and Jena inversions use different fossil-fuel emissions datasets to isolate the net land surface fluxes related to biology. Comparing fossil-fuel emissions for the EU and BA+BNA Arctic and Boreal zones used in each inversion (SI Fig. 1) shows that while the mean emissions were lower in the RIGC inversion, the IAV and trends in absolute fluxes were similar in each inversion. Differences in the fossil emissions are therefore unlikely to contribute significantly to trends in the biological land fluxes of the BA+BNA Arctic and Boreal zones.

## 3.3 Flux amplitude trends

We define the seasonal flux amplitude as the difference between the peak summer $CO_2$ uptake and the maximum $CO_2$ release in the fall, within a calendar year. We examined changes in the flux amplitude using several approaches. Figure 4 shows that the flux amplitude increase, in percent of the mean flux amplitude, is larger in the Arctic zone than the Boreal zone. This is also reflected in the monthly trends in Figure 3cd.

We also examined the change in the seasonal flux amplitude across latitudes from the Jena inversion to see if this observed increase in the seasonal flux amplitude was unique to the high northern latitudes, or if it is more widespread. Here we define the fall flux as the mean of SON, and the summer uptake is fixed as July. Figure 7 shows that significant increases in the annual flux amplitude have occurred between 40–70°N with a peak from 50–65°N. Looking at the fall and summer contributions separately shows that both increasing fall $CO_2$ release and peak summer uptake contribute to the annual amplitude increase, with increasing fall $CO_2$ release outpacing peak summer uptake in the 40–50°N band and increasing peak summer $CO_2$ uptake outpacing fall release in the 55–70°N band (Fig. 7b).

## 3.4 Fire emissions

The net fluxes examined here are dominated by land biosphere fluxes, but they also include $CO_2$ emissions from forest fires, which occur mostly in summer (Fig. S3) (van der Werf et al., 2006). If fire activity were responsible for the trend in summer net carbon uptake, then fire frequency would need to be decreasing. We examined estimates of fire $CO_2$ emissions from 1985 to 2000 from the RETRO compilation and from 1997 to 2012 from the GFEDv4 model (Giglio et al., 2013; Schultz et al., 2008). While we cannot combine the two emissions estimates into a continuous time series because of the different methodologies used, we can examine trends over each record. Neither record shows evidence of decreasing fire emissions over their respective time periods (Fig. S4), therefore, biological activity clearly dominates the trends.

### 3.5    Temperature and NDVI trends

In order to investigate possible drivers of the trends in $CO_2$ fluxes, we also examine trends in surface air temperature and NDVI.  Seasonal temperature changes from 1986 through 2006 were not uniform across the far north (Fig. 8).  In general, warming has been the greatest in the fall (SON) and winter (DJF), although these patterns vary regionally.  Despite the general trend toward warming, cooling trends are seen over the 1986–2006 period for Siberia in winter (DJF) and western North America in spring (MAM). Nearly all of the land regions in the northern hemisphere have experienced warmer summers (JJA) and falls (SON).

Figure 9 shows linear trends from 1986–2006 in gridded NDVI averaged over the "growing season" (April through October) and for the month of July.  Widespread greening trends are observed, with the exception of browning in the southern boreal forest of North America.  Significant greening trends are found in tundra regions, especially in North America.

Figure 10a averages the growing season NDVI over the latitude bands, for each year, showing both the growing season average (top panel) and seasonal maximum (bottom panel).  The period 1986-2006 showed no significant trend in the 50–60°N band in either growing season or maximum metrics (Fig. 9).   In contrast, a significant increasing trend of 0.13% yr$^{-1}$ (Table 2, p=0.01) is found in the >60°N band over this period, driven mostly by trends in tundra regions. Although not included in the trend analysis, growing season NDVI north of 60°N increased abruptly at the end of the record by ~5% from 2009 to 2010. Similarly large changes occurred in 1991–1992 and 1996–1997.

Figure 10b shows the trend in annual peak NDVI, the maximum monthly value for each year regardless of which month it is.  This also shows a small but significant increase north of 60°N, 0.12% yr$^{-1}$ (Table 2,p=0.0103) from 1986-2006 and no significant trend in the 50-60°N region.  Compared to growing season NDVI, the peak NDVI increase from 2009 to 2010 was less extreme.  Plant growth in 2010 was increased in the shoulder seasons in addition to the mid-season peak.

### 3.6    Controls of temperature and NDVI on $CO_2$ fluxes

Summer uptake and fall release of $CO_2$ play a large role in atmospheric $CO_2$ fluxes and concentration amplitude, so here we look at the correlations of $CO_2$ fluxes with temperature and NDVI as proxies for primary productivity and soil respiration variability to help assess mechanistic links.  We performed lagged correlation analysis on monthly time series by calculating the temporal correlation for either the July net $CO_2$ fluxes or fall (SON) fluxes and 3 month running means of temperature or NDVI time series with 0 to 60 month (up to 5 year) lags.  In this analysis, all data sets were de-trended using a stiff spline to remove long-term trends, thus emphasizing processes controlling interannual variability (IAV).  It is also important

to remember that NEE is negative when there is net $CO_2$ uptake from the atmosphere when interpreting the sign of correlations.

Figure 11 shows the lagged correlations for the period 1986–2006 of both inversions for the Boreal zone. We found significantly strong positive correlations between July $CO_2$ NEE and April through August temperatures of the same year, but no evidence of correlation with NDVI. The temperature correlation suggests that warmer growing season temperatures increase soil respiration (or wildfires) and result in reduced peak $CO_2$ uptake (less negative NEE). The RIGC inversion shows significant correlation between warm winters and increased $CO_2$ uptake the following growing season (negative correlation), consistent with Russell and Wallace (2004), but this relation did not appear in the Jena correlation. The fall $CO_2$ fluxes were also positively correlated with growing season temperatures of the same year (not shown). This is consistent with increased growing season air temperatures stimulating soil respiration through the fall, either through increased carbon pools from enhanced summer productivity or warmer soil temperatures that persist into the fall.

Our analysis found significant correlations out to 2 to 4 years prior, suggesting that temperature anomalies could have an impact on NEE after several years delay. While there have been studies suggesting that multi-year lags between climate and $CO_2$ fluxes are important (e.g. Bond-Lamberty et al., 2012), these correlations were not consistent between the inversion models, preventing us for speculating as to the cause.

An analysis of the peak July NDVI correlations with lagged air temperatures showed that warmer temperatures in the 14 months prior to the peak NDVI were associated with higher NDVI in the boreal zone (Fig. 12). One possible interpretation of the correlation analyses presented here is that warmer temperatures in the boreal zone lead to increased plant productivity (indicated by positive NDVI and temperature correlation), but that the IAV of the net C balance in July and the fall was dominated by respiration (indicated by positive NEE and temperature correlation and by the lack of an NDVI and NEE correlation).

Lagged correlations for the arctic zone were generally less significant (Fig. S2). July $CO_2$ fluxes did not show a strong correlation with either temperature or NDVI. Fall $CO_2$ fluxes were not consistently correlated with temperatures in the same season in either inversion. Peak July NDVI was weakly correlated with May-June temperatures. Overall, we conclude that this northern region may be dominated by other controls, like soil thermal processes that would not show up clearly in the correlation analysis with air temperature and NDVI.

**4 Discussion**

The results of these two inversion estimates show that the northern high latitude regions of BNA+BA remain nearly constant or slightly increasing sinks of atmospheric $CO_2$. The Boreal zone, again excluding Europe, absorbed an extra 8–11 Tg C $yr^{-2}$ over the period from 1986 to 2006, resulting in an annual $CO_2$ sink in 2006 that was 170–230 Tg C larger than in 1986. This trend towards increasing $CO_2$ uptake appears to continue through 2012 as indicated by the longer Jena s85 inversion. This result contradicts some modeling studies, which point to trend reversals in observed NDVI and modeled net $CO_2$ fluxes. Hayes et al. (2011) used the TEM ecosystem model to show that increased respiration and fires in this region had weakened the sink strength since 1997. Dynamic global vegetation models have also predicted a trend toward $CO_2$ release to the atmosphere across much of the northern land region from 1990–2009 (Sitch et al., 2015). In general, Carvalhais et al. (2014) found that models tend to over-predict the transfer of carbon from the soils to the atmosphere and overestimate the sensitivity of heterotrophic respiration to climate, although they didn't include the TEM model used by Hayes et al. (2011). The results of this inversion analysis suggest that respiration is over-predicted in models or that increased primary production not captured by these models is offsetting increases in soil respiration and/or forest fire emissions. Our results are also consistent with Forkel et al. (2016) which uses process modeling constrained by atmospheric observations to conclude that photosynthesis has responded more strongly to warming than carbon release processes.

Sensor drift and calibration errors may have resulted in false browning trends in the boreal forest, particularly in the needle-leaf evergreen forests. Recent analyses of NDVI trends in the updated GIMMS3g version find significantly more greening trends than in the previous GIMMSg version observed (Bi et al., 2013; Guay et al., 2014). On a pan-boreal basis, it seems plausible that the $CO_2$ sink strength has continued to increase despite previous reports of drought stress reducing $CO_2$ uptake of the boreal region. As a complication, however, any changes in net carbon fluxes in these ecosystems will depend not only on above ground vegetation changes that can be observed by remote sensing, but also on processes occurring below ground, where most of the carbon is stored (Iversen et al., 2015).

For the Arctic zone, we estimate that July $CO_2$ uptake increased from 1986 to 2006 by 0.15 to 0.27 g C $m^{-2}$ $day^{-1}$, depending on the inversion and the trend detection algorithm. This estimate is based on multiplying the regression slopes (Table 2) by the 21-year time frame. In this zone, we found the strongest NDVI greening trends in the tundra regions, covering roughly 25% of the relevant land area. In light of evidence of rapid shrub expansion in these tundra ecosystems (Myers-Smith et al., 2011; Tape et al., 2006) (Elmendorf et al., 2012), a rapid increase in July $CO_2$ uptake by the tundra ecosystems is plausible.

Most of the previous studies investigating seasonal variability in the northern ecosystem carbon fluxes have relied on observations of atmospheric $CO_2$ concentrations (Angert et al., 2005; Buermann et al., 2013; Keeling et al., 1996; Piao et al., 2008; e.g. Randerson et al., 1999). The analysis presented here is unique in that it considers variability in atmospheric transport through the inversion model approach. Our results are

generally consistent with the finding of Piao et al. (2008) that enhanced $CO_2$ losses from northern ecosystems in the fall partially cancel the enhanced $CO_2$ uptake earlier in the growing season, especially in the arctic zone. We also find evidence of uptake enhancement in the summer as well as the spring in both the arctic and boreal zones, consistent with Graven et al. (2013) and Forkel et al. (2016).

Our investigation of the controls on interannual variability of the $CO_2$ fluxes showed increased $CO_2$ uptake in cooler summers. There are many previous attempts to identify the short-term drivers of the net carbon balance of ecosystems from eddy covariance studies. Several studies have found that warm and dry summers lead to drought stress and reduced net $CO_2$ uptake in boreal forest ecosystems (Arain et al., 2002; McMillan et al., 2008; Welp et al., 2007). Net $CO_2$ flux reductions could be the result of decreased primary
productivity, increased respiration, or both. Correlations between temperature and annual tree ring growth increments point to a switch from a positive correlation to a negative correlation (reduced growth during warm years) driven by increased drought stress in recent decades (Barber et al., 2000; Beck et al., 2011). Wunch et al. (2013) and Schneising et al. (2014) found that summertime total column $CO_2$ in the north was relatively higher during years with warm anomalies in the boreal region, suggesting that reduced net $CO_2$
uptake during warm summers may be driven by the temperature dependence of soil respiration. These correlations of interannual variability are contrary to the overall long-term association with warming and greater summer uptake (Keeling et al., 1996). This difference could reflect the importance of structural ecosystem changes due to warming on the long time scale increasing photosynthesis (Graven et al., 2013), but are also consistent with respiration as the dominant control of NEE on short time scales (Schaefer et al.,
2002). The difference in the time scale of response is important to consider.

A full explanation of the trends in $CO_2$ fluxes of the arctic and boreal zones is still lacking, with possible causes including changes in temperature, which were explored here, but also soil moisture, nutrient status, or fire and insect disturbance. An important unresolved question is how the distribution of deciduous and evergreen plant functional types has changed at the pan-boreal scale over this period. A shift to younger
forests, with increasing deciduous fraction, would increase the seasonal flux amplitude (Welp et al., 2006; Zimov, 1999), perhaps with little change in common NDVI metrics.

The latitude gradient of changes in land fluxes from the Jena inversion, now including results from Europe, shows that, from the 1985–1989 mean to the 2007–2011 mean, the surface flux amplitudes have increased the most from 40°N to 65°N (Fig. 7a). This is consistent with Graven et al. (2013), who argued that, over
the longer period from 1960–2010, the increases in $CO_2$ flux amplitude were centered mostly on boreal regions. Our analysis of the Jena inversion by latitudinal bands shows that the increases in peak July uptake have been greater than the fall $CO_2$ releases north of 55°N, but from 40–50°N, fall release out-paced July uptake (Fig. 7b). The results presented here show that the increased seasonal amplitude in atmospheric $CO_2$ in the high northern latitudes isn't caused by flux trends in the summer or fall only, but
rather both contribute (Fung, 2013; Graven et al., 2013). The trend in annual net $CO_2$ fluxes also includes

changes in other months, with namely greater uptake in the spring (not shown in Fig. 7), which contributes to the annual sum. The advantage of this analysis is that it incorporates interannually varying atmospheric transport, so the temporal changes in the surface fluxes should be better resolved. It does not identify whether individual months have a disproportionately larger influence on the atmospheric $CO_2$ concentration amplitude.

Our attempt to distinguish changes by ~10° latitude arctic and boreal zones is pushing the limits of what is feasible from atmospheric inversions based on sparse atmospheric $CO_2$ observations. The limitation is illustrated by the tendency of the two inverse calculations to allocate the increase in fall $CO_2$ release mostly to different latitude bands and the shift to increasing earlier $CO_2$ uptake in the Jena model compared to the RIGC model in Figure 3c. Resolving fluxes with monthly resolution is also challenging (Broquet et al., 2013), but the long record examined here, by two independent inversions, gives us reasonable confidence in this aspect.

## 5    Conclusions

The two atmospheric inversions analyzed in this study show that the annual net $CO_2$ sink strength in the boreal zone has increased from 1985–2012. However, the annual net $CO_2$ fluxes in the Arctic zone showed no trend. Both regions show significant increases in mid-summer $CO_2$ uptake. But a trend towards greater $CO_2$ emissions in the fall has partly canceled the trend toward greater summer uptake, with the largest cancelation in the Arctic zone. These trends in summer and fall fluxes cause the seasonal amplitude of the fluxes to increase, and consequently, the seasonal amplitude of atmospheric $CO_2$ concentrations.

We also examined NDVI and $CO_2$ flux interannual correlations showing that while warmer summers were correlated with increasing NDVI in the long term, relatively cooler summers favor net $CO_2$ uptake in the boreal region in the short term. This suggests that increased respiration can outpace increases in productivity in the short term. Overall, there is evidence from these atmospheric inversions that increased $CO_2$ uptake from the northern region is offsetting carbon release in the pockets of browning in the boreal zone. Our findings are consistent with the recent NDVI studies from the GIMMS3g product that find overall greening of the boreal and arctic regions (Park et al., 2016; Xu et al., 2013; Zhu et al., 2016). By itself, this would suggest the potential for continued or strengthening net $CO_2$ uptake. However, north of 60°N, our findings show that fall $CO_2$ release largely offsets increased summer uptake in the net annual budget. These results underscore the difficulty of resolving net fluxes from remote sensing indices alone, which can only 'see' the productivity and not the respiration fluxes.

Furthermore, our atmospheric inversions results show no evidence of an overall trend towards increasing $CO_2$ releases in either the boreal or Arctic zone over the 1985-2012 period. This is an important check for process-based biospheric models which tend to predict a shift from sink to source in the next century (Treat

and Frolking, 2013). Our results are not consistent with studies that suggest carbon sinks have weakened in boreal and Arctic ecosystems over past decades (Bradshaw and Warkentin, 2015a; Hayes et al., 2011), but support some process models which do predict an strengthening $CO_2$ sink in the northern region (McGuire et al., 2012). To date, the increase in biomass productivity has appeared to be outpacing $CO_2$ losses from warming northern carbon-rich soils. Time will tell whether this trend continues, or whether it will reverse and become a net carbon source in a few decades as predicted by popular opinion among the community of experts (Abbott et al., 2016).

**Data Availability**

The data used this in this analysis is publically available from the individuals authors responsible for creating the products. The Jena $CO_2$ inversion results are posted to the project website, http://www.bgc-jena.mpg.de/~christian.roedenbeck/download-CO$_2$/.    Run ID s85 version 3.6 was used in this project. Associated files contain the atmospheric monitoring site locations and data used in the inversion and the fossil fuel emissions that were used to solve for the biological land $CO_2$ fluxes. The RIGC $CO_2$ inversion results are posted on the Global Carbon Atlas project website, http://www.globalcarbonatlas.org/?q=en/content/atmospheric-inversions. Likewise, associated files contain the atmospheric monitoring site locations and data used in the inversion and the fossil fuel emissions that were used to solve for the biological land $CO_2$ fluxes. The GIMMS NDVI3g data used is posted on the AVHRR website, https://nex.nasa.gov/nex/projects/1349/. After the final acceptance of this manuscript, the code used in the analysis will be posted on GitHub.

**Acknowledgements**

This project was supported by NASA under award NNX11AF36G, the U.S. Department of Energy under awards DE-SC0005090 and DE-SC0012167, NSF under award PLR-1304270 and UC Multiple Campus Award Number UCSCMCA-14-015. Any opinions, findings, and conclusions or recommendations expressed in this material are those of the authors and do not necessarily reflect the views of the NASA, NSF, DOE, or UC.

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

**Table 1.** $CO_2$ observation stations included in each inversion model.

| Station | Lat | Years | Coverage | RIGC | Jena | Lab |
|---|---|---|---|---|---|---|
| ALT | 82.4 | 1985- 2012 | 95% | X | X | SIO/NOAA |
| BRW | 71.3 | 1985 - 2012 | 100% | X | X | SIO/NOAA |
| STM | 66.0 | 1985 - 2009 | 100% | X | X | NOAA |
| CBA | 55.2 | 1985 - 2012 | 84% | X | X | NOAA |
| SHM | 52.7 | 1985 - 2012 | 88% | X | X | NOAA |
| SCH | 48.0 | 1985 - 2001 | 89% | X | | UBA |
| CMN | 44.1 | 1985 - 2012 | 100% | X | X | NOAA |
| NWR | 40.0 | 1985 - 2012 | 100% | X | X | NOAA |
| RYO | 39.0 | 1987 - 2012 | 90% | X | | NOAA |
| LJO | 32.9 | 1985 - 2012 | | | X | SIO |
| BME | 32.3 | 1989 - 2010 | 81% | X | | NOAA |
| BMW | 32.2 | 1989 - 2012 | 77% | X | | NOAA |
| MID | 28.2 | 1985 -2012 | 97% | X | | NOAA |
| KEY | 25.6 | 1985- 2012 | 95% | X | X | NOAA |
| MLO | 19.5 | 1985 - 2012 | 100% | X | X | SIO/NOAA |
| KUM | 19.5 | 1985 - 2012 | 100% | X | X | SIO/NOAA |
| GMI | 13.4 | 1985 - 2012 | 97% | X | | NOAA |
| RPB | 13.1 | 1987 - 2012 | 84% | X | | NOAA |
| CHR | 1.7 | 1985 - 2012 | | | X | SIO |
| SEY | -4.6 | 1985 - 2012 | 87% | X | | NOAA |
| ASC | -7.9 | 1985 - 2012 | 98% | X | X | NOAA |
| SMO | -14.2 | 1985 - 2012 | 100% | X | X | NOAA |
| AMS | -37.9 | 1985 - 1990 | 89% | X | | NOAA |
| KER | -29.0 | 1985 - 2012 | | | X | SIO |
| CGO | -40.6 | 1985 - 2012 | 99% | X | X | NOAA |
| BHD | -41.4 | 1999 - 2012 | 71% | X | X | NOAA |
| PSA | -64.9 | 1985 - 2012 | 94% | X | X | NOAA |
| SYO | -69.0 | 1986 - 2012 | 78% | X | | NOAA |
| SPO | -89.9 | 1985- 2012 | 97% | X | X | SIO/NOAA |

Coverage refers to percent coverage of observation data from 1985 - 2006 used in the RIGC inversion.
Most station records start before 1985 and continue beyond 2012 but that data was not used in this analysis.
5 Lab stations names and data links as following:
NOAA: NOAA ESRL/CMDL, http://www.esrl.noaa.gov/gmd/ccgg/globalview/co2/co2_observations.html
SIO: Scripps Institution of Oceanography, http://cdiac.ornl.gov/trends/co2/sio-keel-flask/
UBA: Umweltbundesamt and University of Heidelberg, Germany, http://cdiac.ornl.gov/trends/co2/uba/uba-
sc.html

**Table 2.** Trend and significance statistics for time series of interest, from 1986 through 2006. 'Trend' is the slope from linear least squares (LSQ) and Mann-Kendall (M-K) sen slope, likewise, 'Sig (p-value)' is the p-value from LSQ and M-K tests. Arctic zone and boreal zone are for BNA+BA, EU = Europe, NO = northern ocean. Italic values indicate 90% significance level.

| 1986-2006 | | | Trend | | Sig (p-value) | |
| --- | --- | --- | --- | --- | --- | --- |
| **Time series** | **zone** | **inversion** | **LSQ** | **M-K** | **LSQ** | **M-K** |
| $CO_2$ flux annual sum (Pg C $yr^{-2}$) | arctic | RIGC | 0.0350 | 0.0051 | 0.5719 | 0.2389 |
| | arctic | JENA | -0.0028 | -0.0021 | 0.3550 | 0.6077 |
| | *boreal* | *RIGC* | *-0.0110* | *-0.0101* | *0.0724* | *0.0967* |
| | boreal | JENA | -0.0081 | -0.0076 | 0.1438 | 0.1941 |
| | EU | RIGC | 0.0007 | -0.0027 | 0.9295 | 0.7398 |
| | EU | JENA | -0.0052 | 0.0020 | 0.5092 | 0.8326 |
| | *NO* | *RIGC* | *-0.0032* | *-0.0037* | *0.0308* | *0.0320* |
| | *NO* | *JENA* | *-0.0014* | *-0.0015* | *0.0005* | *0.0060* |
| $CO_2$ flux amplitude (% $yr^{-2}$) | *arctic* | *RIGC* | *0.93* | *0.85* | *0.0019* | *0.0201* |
| | *arctic* | *JENA* | *1.04* | *0.97* | *0.0002* | *0.0004* |
| | boreal | RIGC | 0.15 | 0.22 | 0.3241 | 0.2639 |
| | *boreal* | *JENA* | *0.44* | *0.39* | *0.0328* | *0.0372* |
| | *EU* | *RIGC* | *0.62* | *0.55* | *0.0769* | *0.1390* |
| | EU | JENA | 0.18 | 0.16 | 0.2723 | 0.3812 |
| | *NO* | *RIGC* | *-2.35* | *-2.20* | *0.0192* | *0.0372* |
| | NO | JENA | 0.63 | 0.65 | 0.2894 | 0.2639 |
| $CO_2$ flux July (g C $m^{-2}$ $day^{-1}$ $yr^{-2}$) | *arctic* | *RIGC* | *-0.0128* | *-0.0120* | *0.0167* | *0.0655* |
| | *arctic* | *JENA* | *-0.0072* | *-0.0082* | *0.0004* | *0.0028* |
| | boreal | RIGC | -0.0058 | -0.0034 | 0.1712 | 0.3492 |
| | *boreal* | *JENA* | *-0.0097* | *-0.0085* | *0.0615* | *0.0571* |
| Fossil fuel emissions (Pg C $yr^{-2}$) | *arctic* | *RIGC* | *-0.0206* | *-0.0011* | *0.0024* | *0.0320* |
| | arctic | JENA | -0.0159 | 0.0001 | 0.9664 | 0.9759 |
| | *EU* | *RIGC* | *-0.0013* | *-0.0195* | *<0.0001* | *0.0002* |
| | *EU* | *JENA* | *-0.0000* | *-0.0135* | *<0.0001* | *0.0086* |
| NDVI gs (% $yr^{-2}$) | *arctic* | | *0.1500* | *0.1300* | *0.0112* | *0.0103* |
| | boreal | | 0.0587 | 0.0532 | 0.3170 | 0.4503 |
| NDVI peak (% $yr^{-2}$) | *arctic* | | *0.1200* | *0.1200* | *0.0109* | *0.0103* |
| | boreal | | 0.0030 | -0.0043 | 0.9419 | 0.9759 |

| Time series | zone | | LSQ | M-K | LSQ | M-K |
|---|---|---|---|---|---|---|
| Spring T (°C yr$^{-2}$) | arctic | | -0.0124 | 0.0439 | 0.8592 | 0.7858 |
| | boreal | | 0.0063 | 0.0192 | 0.8961 | 0.6506 |
| Summer T (°C yr$^{-2}$) | *arctic* | | *0.0848* | *0.0915* | *0.0011* | *0.0041* |
| | *boreal* | | *0.0491* | *0.0527* | *0.0043* | *0.0072* |
| Fall T (°C yr$^{-2}$) | *arctic* | | *0.0500* | *0.0377* | *0.1559* | *0.0655* |
| | *boreal* | | *0.0551* | *0.0503* | *0.0080* | *0.0072* |
| Winter T (°C yr$^{-2}$) | arctic | | -0.0159 | -0.0096 | 0.6961 | 0.6077 |
| | boreal | | -0.0041 | 0.0122 | 0.9172 | 0.8326 |

**Table 3:** Same as Table 2, but for the period from 1985 through 2012.

| 1985-2012 | | | Trend | | Sig (p-value) | |
|---|---|---|---|---|---|---|
| Time series | zone | inversion | LSQ | M-K | LSQ | M-K |
| $CO_2$ flux net annual (Pg C yr$^{-2}$) | *arctic* | *JENA* | *-0.0040* | *-0.0038* | *0.0411* | *0.0722* |
| | *boreal* | *JENA* | *-0.0072* | *-0.0078* | *0.0655* | *0.1095* |
| | EU | JENA | -0.0032 | 0.0005 | 0.5155 | 0.9842 |
| | NO | JENA | -0.0010 | -0.0010 | 0.0001 | 0.0037 |
| $CO_2$ flux amplitude (% yr$^{-2}$) | *arctic* | *JENA* | *0.81* | *0.85* | *<0.0001* | *0.0001* |
| | *boreal* | *JENA* | *0.35* | *0.31* | *0.0094* | *0.0187* |
| | EU | JENA | <0.01 | <0.01 | 0.4365 | 0.4179 |
| | NO | JENA | <0.01 | <0.01 | 0.4299 | 0.3740 |
| $CO_2$ flux July (g C m$^{-2}$ day$^{-1}$ yr$^{-2}$) | *arctic* | *JENA* | *-0.0068* | *-0.0072* | *<0.0001* | *0.0001* |
| | *boreal* | *JENA* | *-0.0083* | *-0.0084* | *0.0345* | *0.0380* |

**Figure captions**

**Figure 1:** The major land and ocean basis regions used in the RIGC inversion based on the TransCom3 regions. The Jena inversion was done on a ~4x5 degree grid and aggregated to these regions. The northernmost land regions are shown in color. The two zones that we discuss in this analysis cover Boreal North America and Boreal Asia and are marked in shades of blue, with the arctic zone (>60°N) in light blue and the boreal zone (50°N to 60°N) in dark blue. The European basis region, in red, is not divided at 60°N in the RIGC inversion and therefore is not included in this analysis. Stippling indicates the boreal forest biome based on the GLDAS UMD modified IGBP land classification scheme (http://ldas.gsfc.nasa.gov/gldas/GLDASvegetation.php). The tundra biome is north of the stippling.

**Figure 2:** Annual $CO_2$ fluxes normalized by subtracting the 1986–2006 mean value for (a) arctic zone (>60°N) and (b) boreal zone (50°N to 60°N). Black shows the RIGC inversion results. Grey shows the Jena s85 inversion results. Dashed lines are linear trends from 1986 to 2006. Negative values represent uptake of $CO_2$ by the land biosphere, i.e. out of the atmosphere (Table 2).

**Figure 3:** Mean monthly $CO_2$ fluxes for (a) arctic zone (>60°N) and (b) boreal zone (50°N to 60°N). Black circles are the 1986–2006 means of the RIGC inversion. Grey squares are the Jena s85 inversion over that same period. Magenta is the Jena inversion average over a longer time period (1985–2012). Differences are likely due to differences in atmospheric transport, including vertical mixing, between the models. Linear monthly trends of (c) arctic zone and (b) boreal zone for the same inversions and time periods as in (a) and (b) (Table 2).

**Figure 4:** $CO_2$ flux amplitude for each year calculated as the maximum monthly flux (positive = $CO_2$ release to the atmosphere) minus the minimum monthly flux (negative = $CO_2$ uptake by the biosphere) for (a) arctic zone and (b) boreal zone. Black shows the RIGC inversion, grey shows the Jena s85 inversion. Dashed lines show the linear trends from 1986–2006, the common period between the inversions.

**Figure 5:** July $CO_2$ flux for each region and inversion normalized by subtracting the 1986–2006 mean value. This is the month of maximum $CO_2$ uptake in each case (see Figure 3). The dashed lines are the linear trends from 1986 to 2006, also plotted in Figure 3c and d.

**Figure 6:** Fluxes from the Northern Ocean and European basis regions. (a) Annual fluxes and (b) annual flux amplitude for the Northern Ocean. (c) Annual sum and annual flux amplitude for the European region. The trends in these fluxes are small, and in the case of Europe, in the same direction, compared to the trends resolved for the Boreal North America and Boreal Asia regions.

**Figure 7:** Latitudinal gradients in the land $CO_2$ fluxes from the Jena s85 inversion. (a) Green is the difference from the 2007–2011 mean from the 1983–1989 mean in the July $CO_2$ uptake with the sign reversed (here positive is uptake by the biosphere) and magenta is the difference in the mean of Sep–Nov fall $CO_2$ release with conventional sign (positive is release of $CO_2$ to the atmosphere). (b) The difference between the 2 curves in (a) showing the change in $CO_2$ seasonal flux amplitude in Pg C yr$^{-1}$. Positive values reflect an increase in the peak-to-trough flux amplitude.

**Figure 8:** Gridded temporal trends in surface air temperature from the GISS temperature record (data.giss.nasa.gov). Plots were made using software available on the data archive website.

**Figure 9:** Gridded temporal trends in GIMMS 3G NDVI (a) growing season (Apr–Oct) mean and (b) July only from 1986–2006. Trends are expressed as percent changes from the mean.

**Figure 10:** Time series of NDVI trends averaged for the analysis regions in this study. (a) growing season (Apr–Oct) mean and (b) annual maximum, usually in July. Black is the arctic zone and grey is for the boreal zone.

**Figure 11:** Correlation coefficients for July $CO_2$ fluxes in a given year (Year 0) from the boreal zone with lagged 3-month running mean temperature (area-weighted and NPP-weighted) and NDVI for the same region. (a) RIGC inversion and (b) Jena s85 inversion over the current and previous 4 years. Positive correlations mean that high temperature or NDVI leads to less $CO_2$ uptake. Filled circles indicate significance greater than the 95% level. Shaded bars indicate the summer months (May–August).

**Figure 12:** Correlation coefficients for maximum NDVI in given year (Year 0) with lagged 3-month running mean temperature (area-weighted and NPP-weighted). Positive correlations mean greater NDVI during (or following) warmer temperature. Filled circles indicate significance greater than the 95% level. Shaded bars indicate the summer months (May–August).

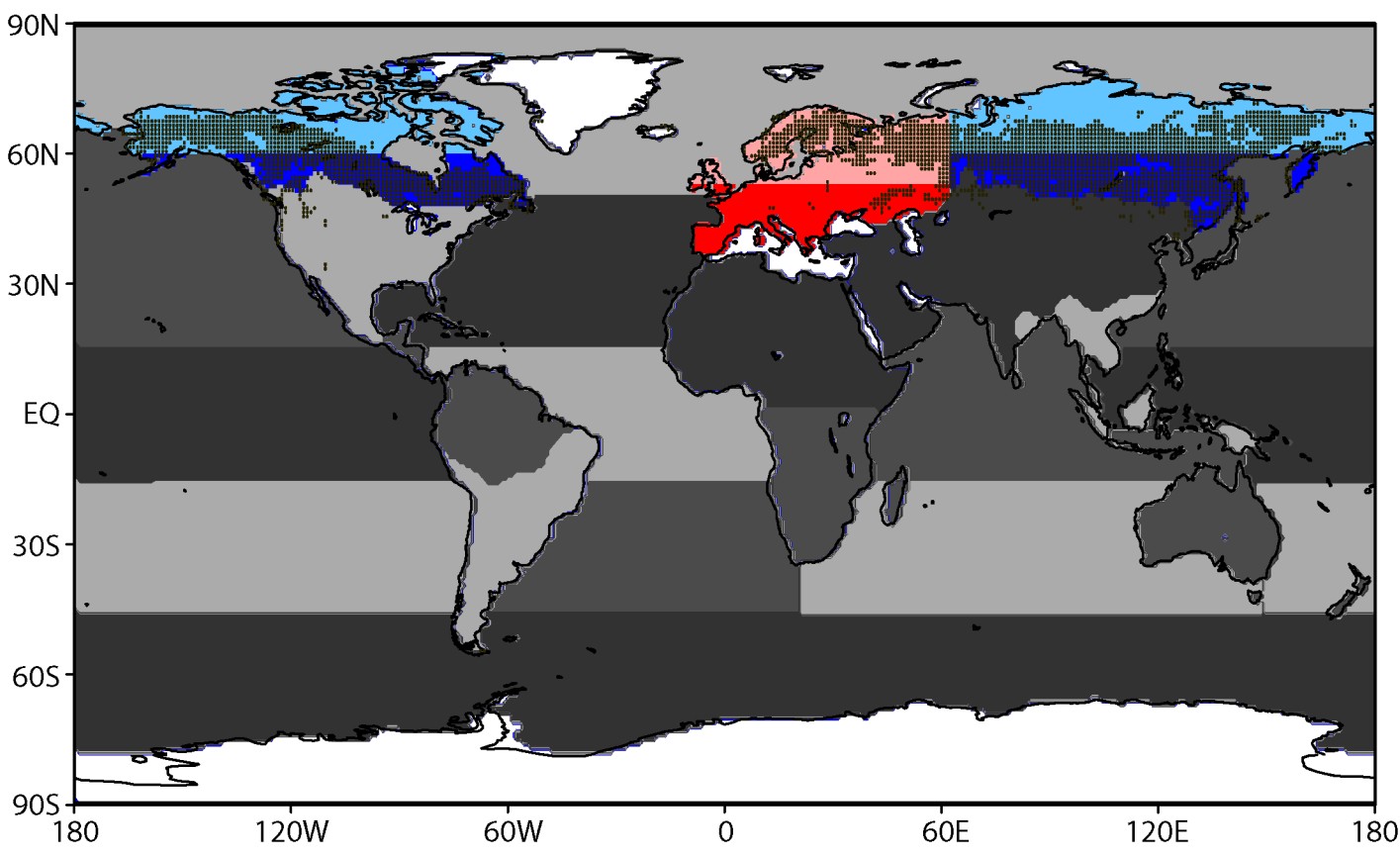

Figure 1

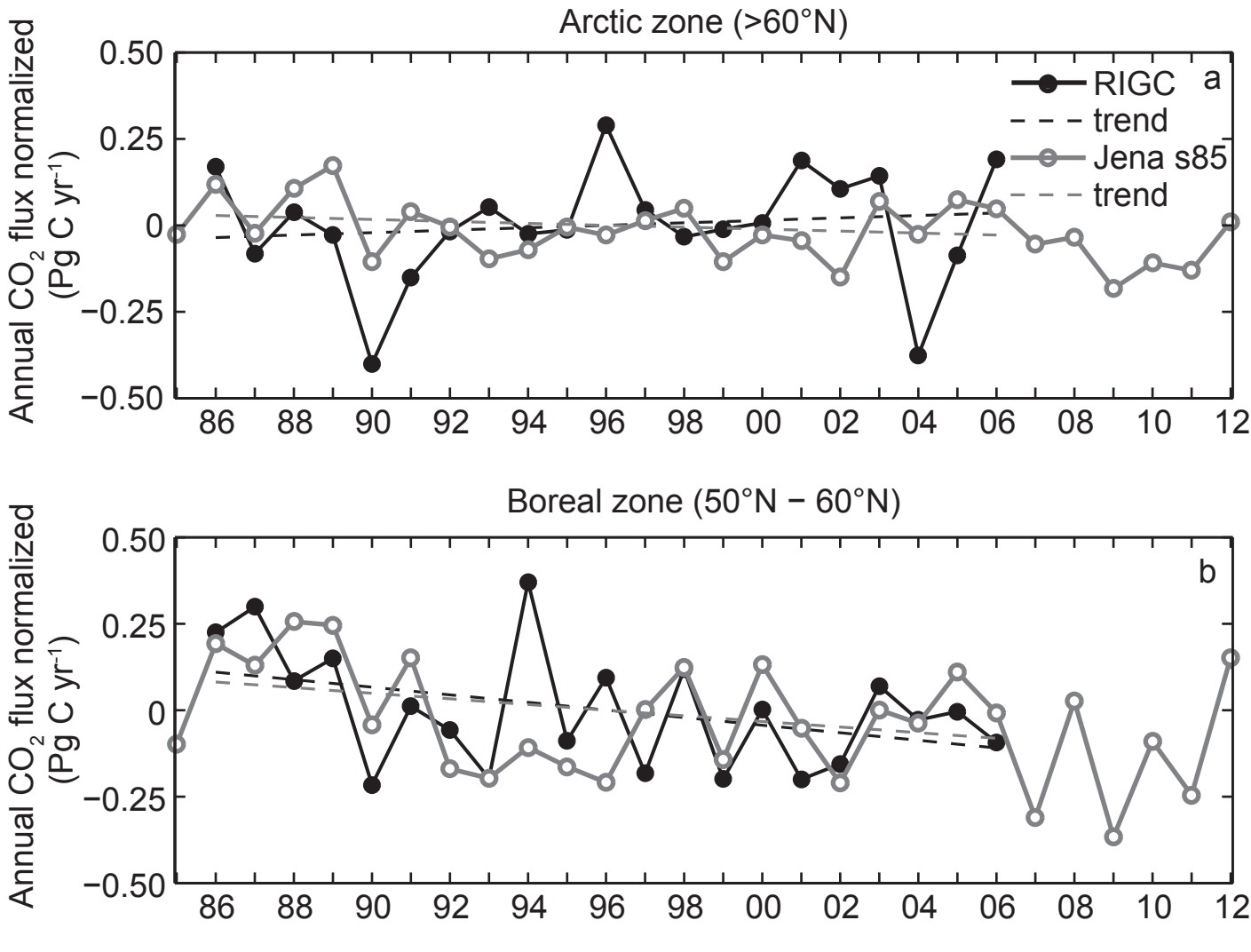

**Figure 2**

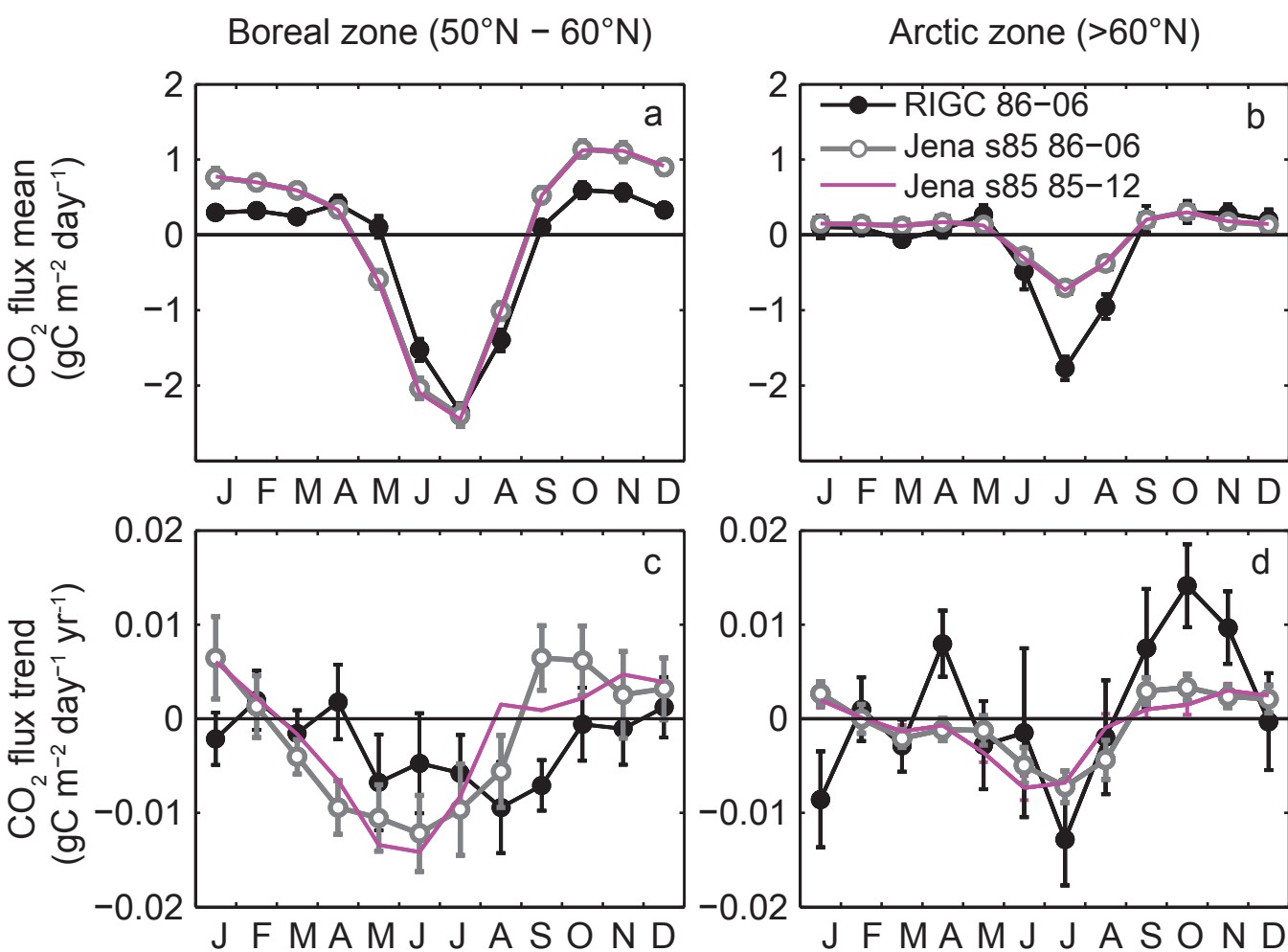

**Figure 3**

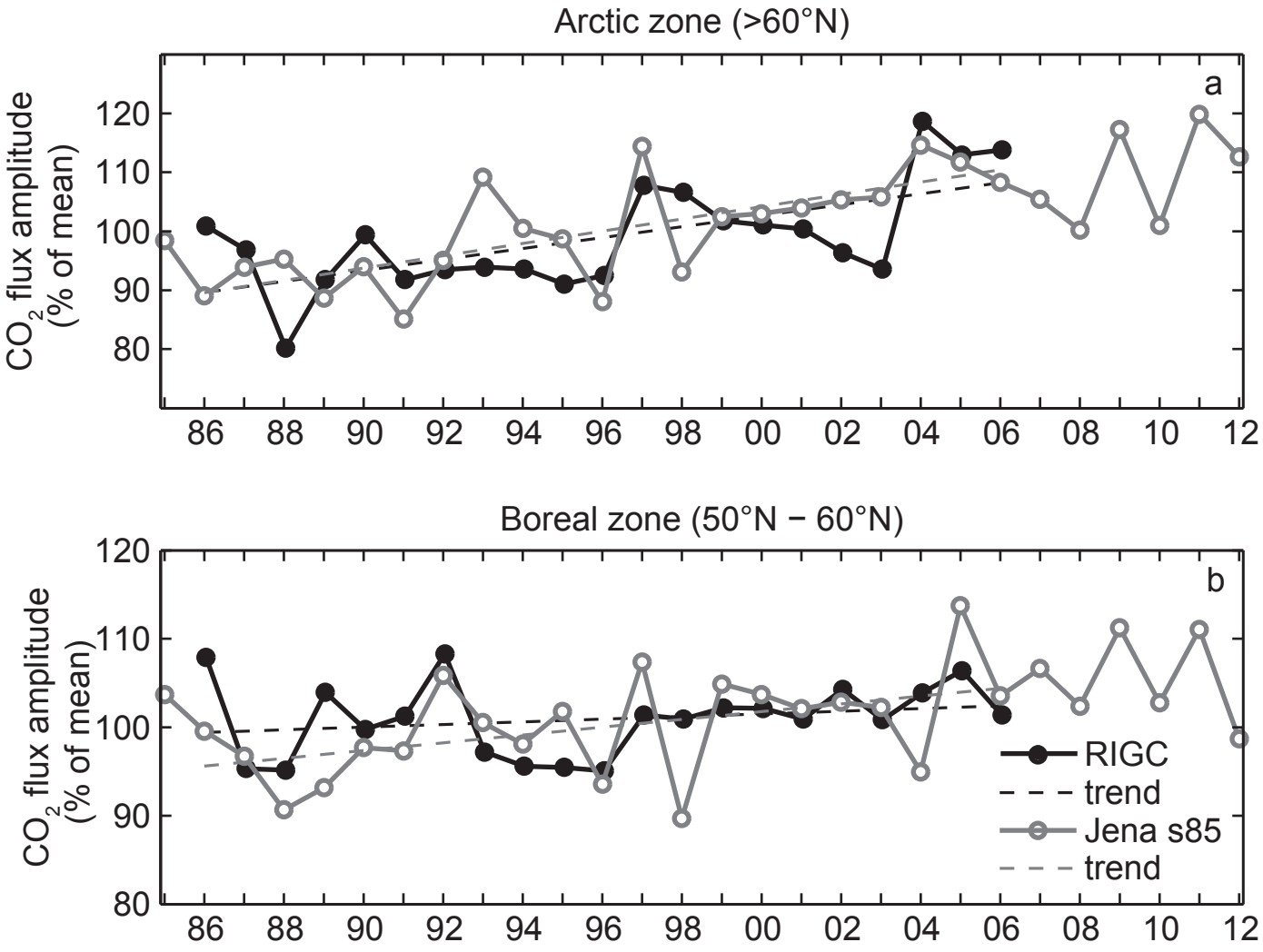

**Figure 4**

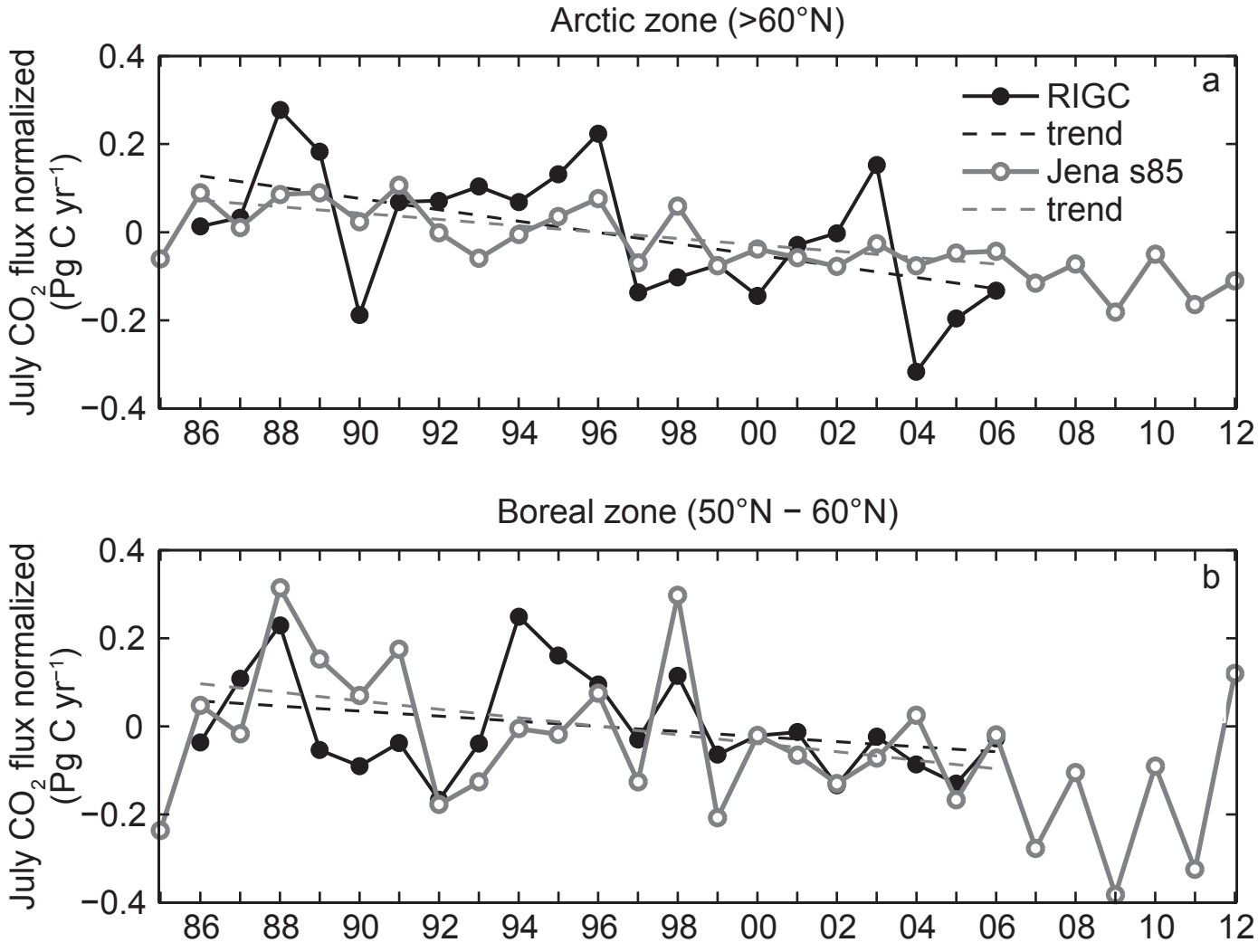

**Figure 5**

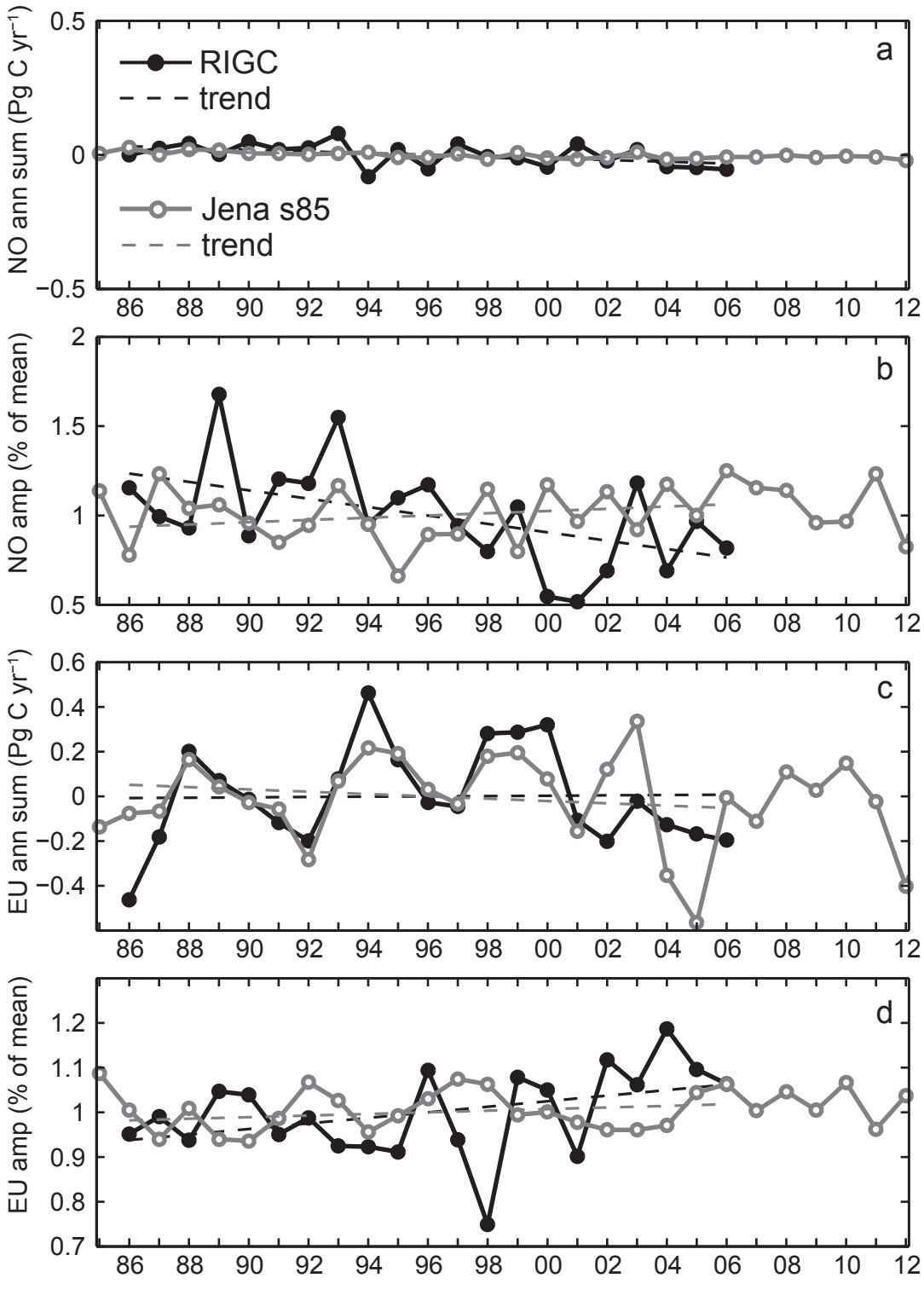

**Figure 6**

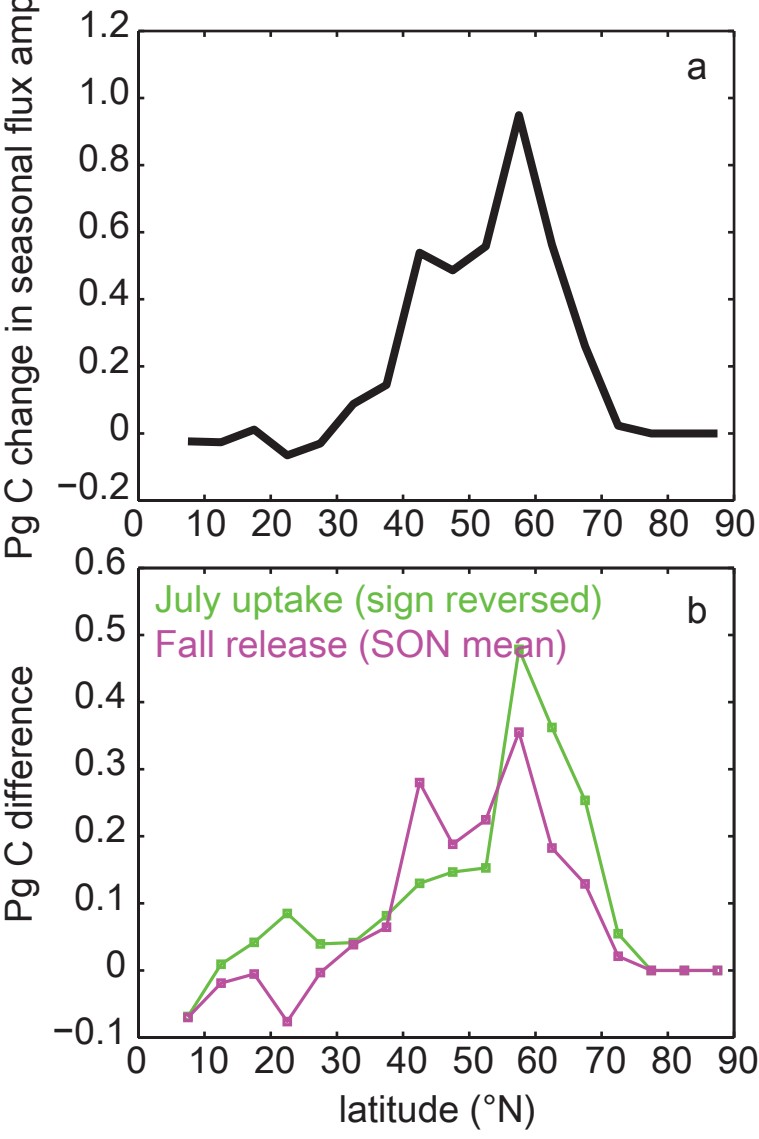

**Figure 7**

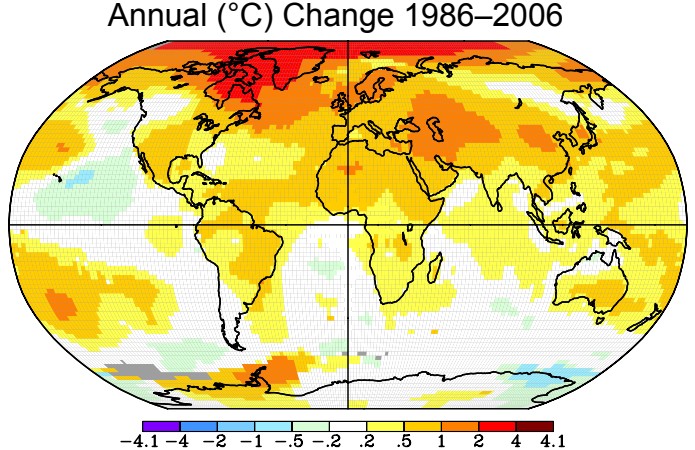

**Figure 8**

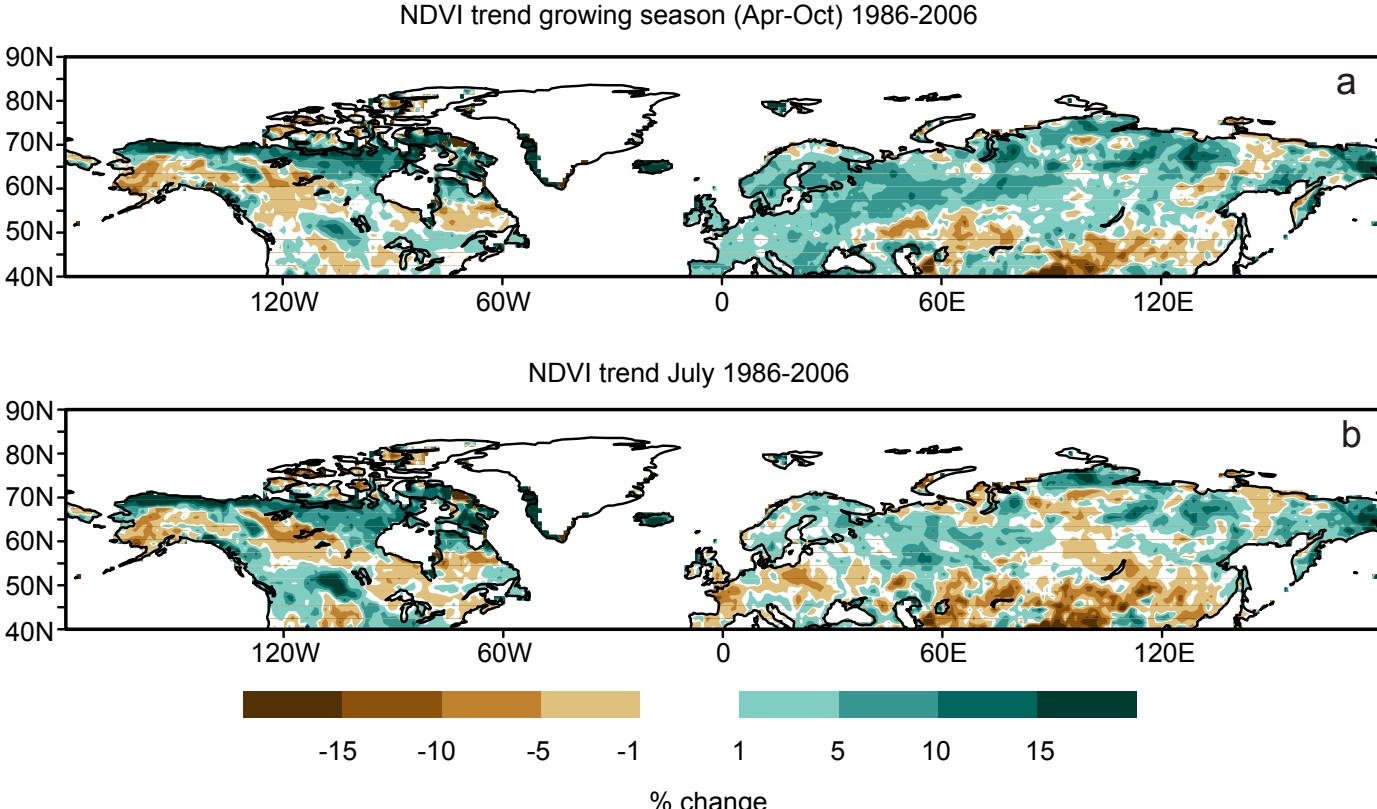

**Figure 9**

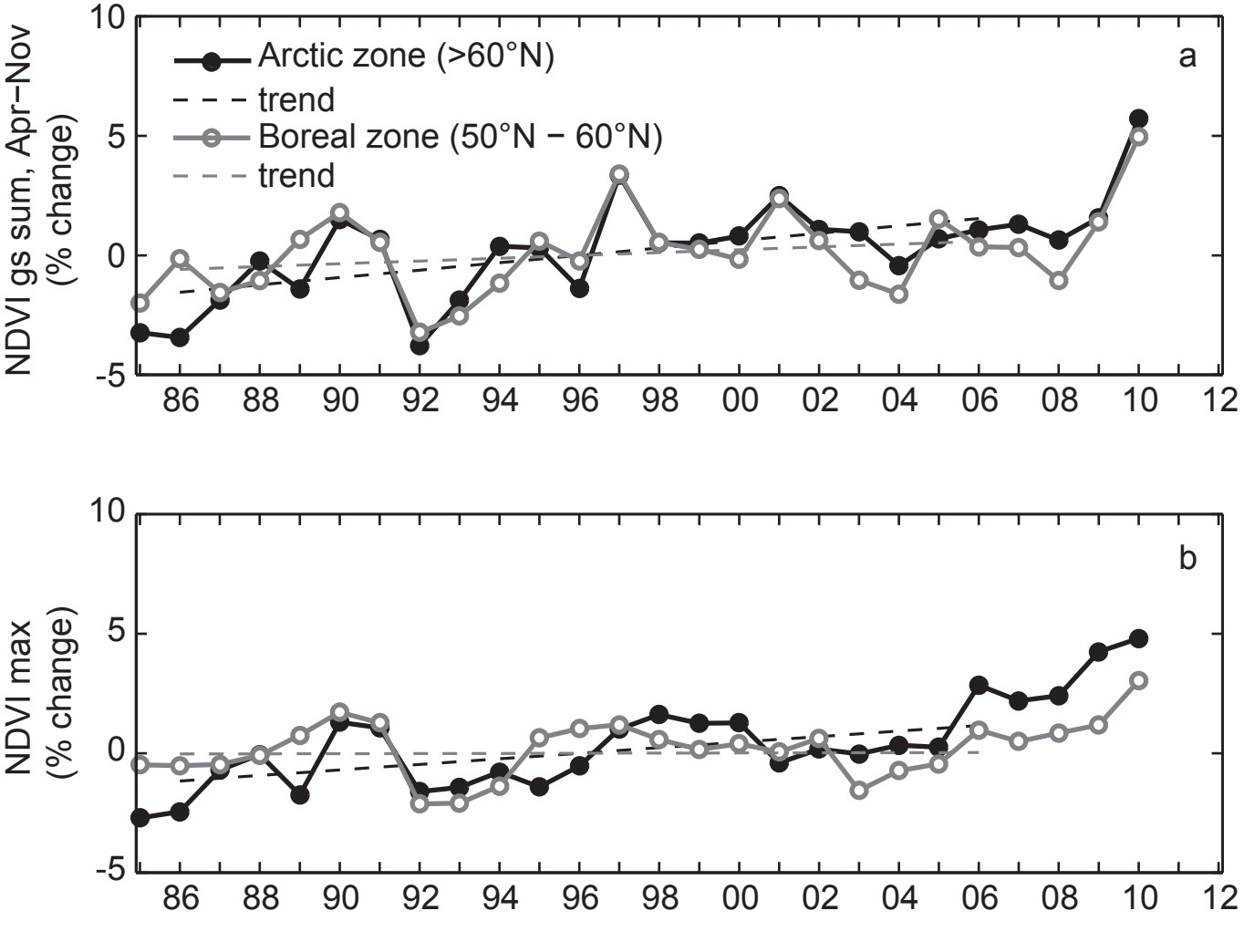

**Figure 10**

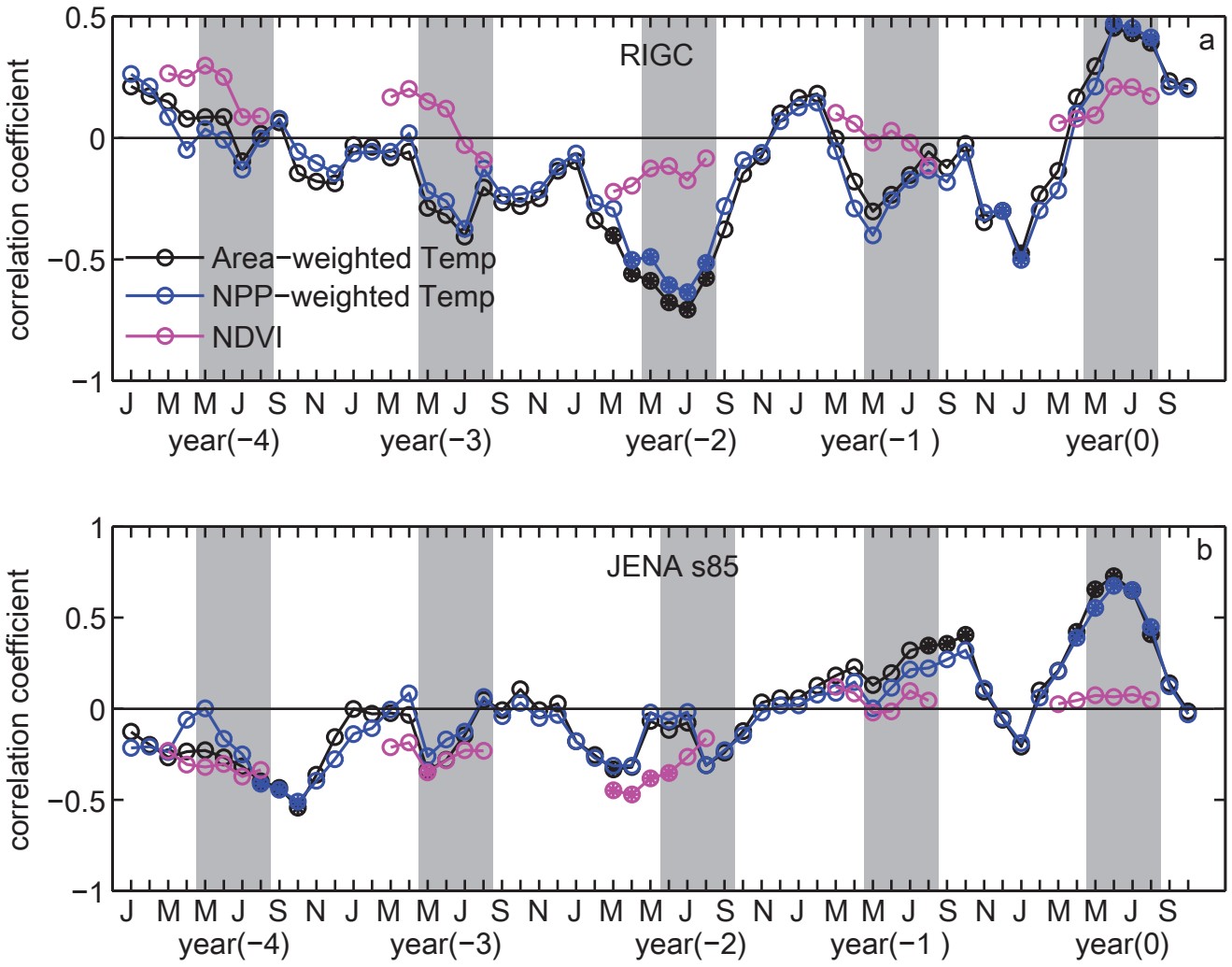

**Figure 11**

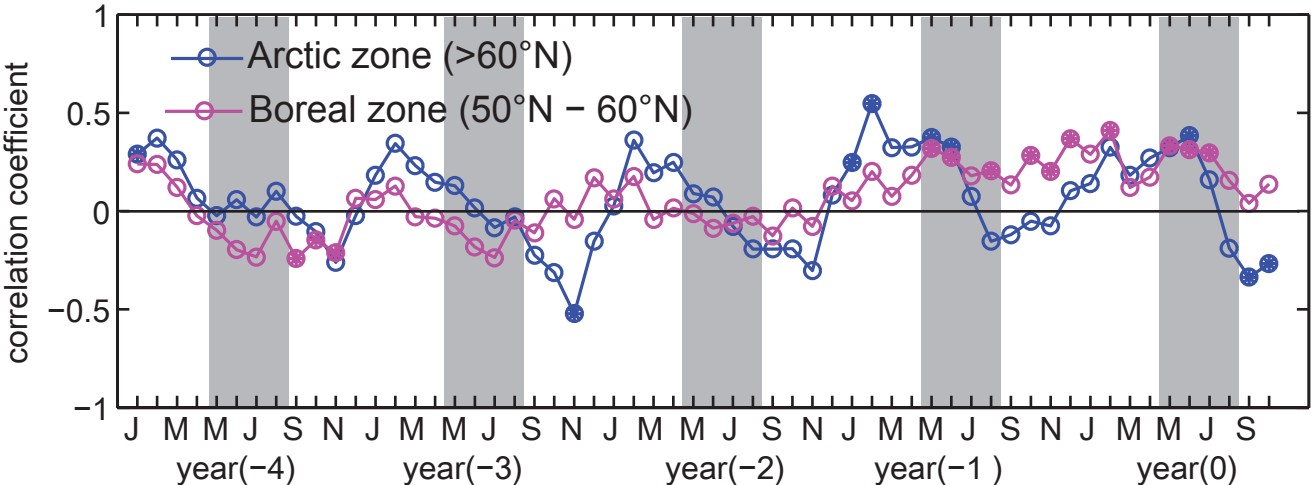

**Figure 12**