# Peer review of "Increasing summer net CO2 uptake in high northern ecosystems inferred from atmospheric inversions and comparisons to remote sensing NDVI"

_Atmospheric Chemistry and Physics, 2016_

## Referee Comment (RC1) · Anonymous Referee #3 · 17 Mar 2016

This paper uses the results from two atmospheric inversion models and long-term surface temperature and NDVI records to compute trends in the CO2 fluxes in the Arctic and Boreal regions (excluding Europe). The authors conclude that the Boreal region has become an increasingly large sink for CO2, with no statistically significant change in the Arctic, even though the seasonal cycle amplitude in CO2 in both regions has increased. The authors argue that this is due to the balance between increased summertime uptake and fall CO2 emissions.

The paper is well-written and clear, and suitable for publication in ACP. I recommend that this paper is published after addressing the following comments.

Main Comments:

I would recommend that the authors look at the more recent solar-induced fluorescence (SIF) measurements (e.g., GOME-2, GOSAT, OCO-2) in their analyses. SIF is reported to be more directly related to photosynthesis than greenness indices are, and show some significant differences in the Boreal and Arctic regions (e.g., Joiner et al. 2013). GOME-2 has the longest time series (launched in 2006), and I recognise that this does not cover the main time period of the inversions, but it should be helpful to determine whether NDVI is fully capturing the productivity cycle in the Boreal region.

This analysis does not directly consider the timing of the onset of the growing season, but it is obvious in Figure 3a that even between the two models using the same CO2 concentration data, the phase and duration of the growing season are inconsistent. This raises several questions: Are monthly fluxes temporally fine enough for this analysis (i.e., would the results change if you were to look at, say, bi-weekly fluxes)? Do the two inversions show a similar change in the timing of the onset of the growing season over time? Do they show consistent changes in the length of the growing season?

Minor Comments:

Title: I suggest you clarify the title by specifying that the inversions use surface concentrations and that the remote sensing is of NDVI and temperature P2L22: ... trigger *a* massive... P2L35: Is (1997) referring to a paper? P5L2: Be careful to state that GLOBALVIEW-CO2 isn't "data". From the ESRL webpage (http://www.esrl.noaa.gov/gmd/ccgg/globalview/co2/co2_intro.html): "GLOBALVIEW-CO2 is derived from atmospheric measurements but contains no actual data." P6Para24: Please clarify. I find the first two sentences very confusing. P9L25: In order *to* investigate... P10L6: You show the average growing season NDVI. Would the integrated NDVI over the growing season be better correlated with CO2 uptake? P10L13: How does the month of the maximum NDVI change over time? Is there a trend? P10L27: How is significance defined here? P12L28: ... warm summers may *be* driven... P12L27: Schneising et al. (2014) also came to a similar conclusion. P13L22: ... to different *latitude* bands... Figure 3: The two inversions differ in their

mean seasonal cycle amplitudes by a factor of two in the Arctic, and they have significantly different onsets of the growing season in the Boreal zone. Can you explain why?

References:

Joiner, J., Guanter, L., Lindstrot, R., Voigt, M., Vasilkov, A. P., Middleton, E. M., Huemmrich, K. F., Yoshida, Y., and Frankenberg, C.: Global monitoring of terrestrial chlorophyll fluorescence from moderate-spectral-resolution near-infrared satellite measurements: methodology, simulations, and application to GOME-2, Atmos. Meas. Tech., 6, 2803-2823, doi:10.5194/amt-6-2803-2013, 2013.

Schneising, O., M. Reuter, M. Buchwitz, J. Heymann, H. Bovensmann, and J. P. Burrows (2014), Terrestrial carbon sink observed from space: variation of growth rates and seasonal cycle amplitudes in response to interannual surface temperature variability, Atmos. Chem. Phys., 14(1), 133–141, doi:10.5194/acp-14-133-2014.

---

## Referee Comment (RC2) · Anonymous Referee #2 · 25 Mar 2016

This study takes an important step beyond the well-documented increase in atmospheric CO2 seasonal amplitude at Arctic monitoring sites and asks whether this amplitude increase actually reflects a net gain in CO2 uptake in boreal and Arctic regions. The methodology involves exploring trends in NEE fluxes inferred from 2 different inversion systems over the common period 1986-2006 (and also 1985-2012 for one of the inversions). In general, the study is well presented and documented and I support publication with minor revision. Some of my more important (although still relatively minor) concerns are that the results differ substantially between the two inversions in many aspects, leading to doubts about the robustness of either. Also, the Arctic zone >60N is the region with the most unequivocal increase in CO2 amplitude, yet the inver-

sions estimate significant trends in net CO2 uptake mainly in the boreal zone (50-60N), not the Arctic zone. The CO2 amplitude increase at Barrow, AK (71N) in particular has been the subject of much attention, yet it doesn't seem to be associated with an actual increase in CO2 uptake in the surrounding region. A particularly interesting result is that the inversions suggest that increased CO2 respiration and release in fall may largely balance increased CO2 uptake in summer (although they don't agree where the increased fall respiration is occurring). I am curious about the heavy focus on midsummer (July) at the expense of late spring/early summer, when the CO2 cycle (e.g., at Barrow) indicates an earlier onset of photosynthesis. Could this be when some of the net gain in CO2 uptake is occurring? Re: the 2 time periods chosen: 1986-2006 and 1985-2012. I suggest making the second period 1986-2012, to remove ambiguity about why the results differ between the 2 periods. With the 1985 start year, we don't know whether the changes in the trends are due to the influence of starting in 1985 vs. 1986 or due to more recent changes from 2006-2012. The latter possibility seems more relevant to global change, therefore I suggest eliminating this ambiguity by starting both periods in 1986. Trend calculations of this sort can be sensitive to the starting year, especially when operating on the margins of statistical significance, as is the case here. On a related note, is the legend in Fig 3b (86-12) a typo?

Some specific comments Abstract, there are a couple of grammatical errors or typos that interfere with smooth reading: AbL17-18 "Here we examine CO2 fluxes from northern boreal and tundra from 1986 to 2012 ..." AbL29-31 sentence beginning with "Meanwhile . . . P2L35 (1997)? P3L20-26 Please define what exactly is meant by "browning" and "greening," e.g., does this refer to changes in seasonality of NDVI, or does it refer to an annual mean index? The Introduction in general is quite good and informative, but is marred by the paragraph on L8-17. I have several suggestions for improving it: P4 L8-17 The emphasis on aboveground vs. belowground in the first sentence seems incongruous because it is not mentioned earlier as a strength of inversions. Perhaps start this paragraph with a more general statement about the strengths of forest inventories. P4 L13 For clarity, should "several studies" be "several

process-based model studies"? P4 L11-17 Can we believe these results? What are the weaknesses of process-based model studies? (Referring back to earlier statement that, "Each of these methods has its strengths and weaknesses.") P4L18 "...50N, using the atmospheric inversion method." P4L35 What is "It" ? P5L7 What period? P5L9 Temporal coverage of what? Years, months, weeks? What is the time resolution? P5L21 What is LPJ? P5L30 What are the units of NDVI? Are they mass units, e.g., kg/m2 or flux units, e.g., in kg/m2/s? P6L10- Perhaps I am missing something, but I don't see the 2 different analysis methods for trends and significance reported in Table 2 described anywhere in this section. There is only a brief mention of them in the Table 2 caption, which is not very informative. P6L15 In Figure 1 the boreal forest stippling extends well north of 60 degrees. Does this mean that the so-called Arctic zone consists largely of boreal forest? This is somewhat confusing and perhaps should be noted here. Other parts of the text seem to suggest the Arctic zone is mainly tundra, but later p.12 mentions that tundra covers only 25% of the Arctic zone. Figure 3c,d. Should the Y-axis units be gC/m2/day per year? P7L33. Probably should note that P < 0.1 is significant at only 10% level, which is a weak standard. In general p < 0.05 is the standard level required for significance. P9L13. How were these 40-50 and 55-65N bands chosen? Figure 7 seems to suggest net release and net uptake for 40-55N and 55-75N, respectively. Also, please check P13L10 for consistency. P9L25 In order to ... P10L27 "We found significantly strong positive correlations between July CO2 flux and April through August temperatures of the same year..." The next sentence is confusing because it suggests lower CO2 uptake (more release) in warm years, in contrast to the quoted sentence – please clarify that "positive correlation" means the July flux is weaker not stronger. P12L17-20 "Increased summer CO2 uptake cannot be explained by earlier spring leaf-out, but rather points to changes in mid-summer photosynthetic and respiration fluxes themselves." Where is this sentence supported in the Results? P12L31-33 "This difference could reflect the importance of structural ecosystem changes due to warming on the long time scale increasing photosynthesis (Graven et al., 2013), but on the short time scale, respiration is the

dominant control." This seems like a core conundrum of this study (together with the fact that no apparent increase in net $CO_2$ uptake is occurring in the band where the $CO_2$ amplitude is increasing). Both of these points might be worth discussing more.

Please also note the supplement to this comment:
http://www.atmos-chem-phys-discuss.net/acp-2016-10/acp-2016-10-RC2-supplement.pdf

---

## Referee Comment (RC3) · Anonymous Referee #1 · 29 Mar 2016

This paper is competently written, and I don't find obvious errors in method, analysis, or results. My main complaint has to do with context and integration of results into previous results.

The authors spend some time reviewing previous, sometimes contradictory studies of the boreal and arctic regions

- conflicting browning/greening NDVI studies

- the 'carbon bomb' vs. the authors' results that don't show a large carbon efflux from permafrost regions

- high northern latitudes have decreasing sink, or even becoming a net source vs. the

present study that disagrees with this result

After multiple readings of the paper, I'm not sure how far this work goes towards resolving any of these questions, but I think potential is there to do so.

The basic result, that there is increasing $CO_2$ uptake in the boreal region (not in the arctic) while the amplitude of arctic $CO_2$ cycles has increased, seems reasonably well-established by the results of their study. What I don't really get is a sense of how these results fit into the literature to confirm or deny other hypotheses as a means to clarify our understanding of this admittedly complex region.

In the introduction the authors say that "The net carbon balance of increased plant growth and increase soil respiration is unclear, but has important consequences for predicting carbon-climate feedbacks." By the end of the paper, I don't get the feeling that the authors make a definitive statement addressing this one way or the other.

I believe this study has merit, and that any flaws are not fatal. A more rigorous organization of previous literature and the place of this study within our understanding would be helpful. Also, it seems that perhaps the authors are being too passive and 'nice' here, and are just presenting their results without directly confirming or refuting the work of others. Be bold! In the conclusion, state who among your predecessors you agree with, who you disagree with, and say why. You take the risk of perhaps ruffling a few feathers, but you will ensure response, and that's a very effective way to move science forward. (I'm reminded of a current disagreement between a group that hypothesizes that the Amazon experiences greenup during drought, and the group that believes this isn't the case. The issue has not been resolved, but there have been some very interesting studies that have come out of the dispute.)

Some specific comments:

- Author is not listed in reference in the 4th paragraph of the introduction.

- The Jena inversion uses LPJ land flux and Mikaloff Fletcher/Takahashi ocean flux.

What does the RIGC inversion use? How are these surface fluxes similar/different, what might that mean for inversion results? Could these differences be the source of the RIGC peak $CO_2$ uptake being double that of Jena (section 3.1.1)?

- Section 3.3: the authors claim that the flux amplitude increase, shown in figures 3cd, is larger in the arctic than in the boreal regions. This is clearly true in the RIGC product, especially with regard to SON efflux. However, I'm not sure I agree that this is true for Jena. To my (subjective) eye, the summer uptake and fall/winter efflux amplitude increase is larger for both Jena products in the boreal region than in the arctic.

- I'm a bit confused about the results shown in sections 3.5 and 3.6, Figure 11. Figure 3 clearly displays a strong amplification of July $CO_2$ uptake, and Figure 8 shows a clear upward trend in JJA temperatures over the period of study. But Figure 11 (and references to studies in the text) correlate cooler summertime temperatures with increased uptake. What am I missing here? These seem contradictory. Is the moisture component the more important than the temperature?

- Section 3.5: Russell and Wallace (2004) and Schaefer et al. (2002) looked at carbon flux in relation to modes of climate variability such as the annular modes. Hurrell et al. (2001) discussed trends in the NAO itself. Would studies such as these help provide context here, or are they unrelated?

- Is the last paragraph of section 3.6 necessary?

- Figure S1: RIGC BA+BNA fossil fuel (ORNL/EDGAR) is about half the Jena anthropogenic flux for the same region (also EDGAR, but apparently different version. Intuitively, I would expect that Jena uptake would have to be larger than RIGC to resolve observed $CO_2$ concentration with these anthropogenic fluxes. Why isn't this the case?

- Patra et al. (2008) and Parazoo et al. (2008) discuss model resolution in relation to simulations of $CO_2$. I wonder if advection of the effect of large surface $CO_2$ flux into

boreal/arctic regions is a partial (or dominant?) cause of the increasing amplitude of high-latitude CO2 concentrations? Or is Graven et al. (2013) the last word? What role might model resolution play? Are these issues not germane to this manuscript?

- Figure 11: There are significant correlations out to two years for RIGC and 4 years for Jena that are not discussed in the text. What might these long time-lag correlations mean?

References

Hurrell, J.W., Y. Kushnir, M. Visbeck 2001: The north atlantic oscillation. Science, Vol. 291, No. 5504 (Jan 26 2001), p603-605.

Parazoo, N.C., A.S. Denning, S.R. Kawa et al., 2008: Mechanisms for synoptic variations of atmospheric CO2 in North America, South America, and Europ. Atmos. Chem. Phys., 8, 7239-7254.

Patra, P.K., R.M. Law, W. Peters, et al., 2008: TransCom model simulation of hourly atmospheric CO2: Analysis of synoptic-scale variation for the period 2002-2003. Glob. Biogeochem. Cy., 22, GB4013, doi:10.1029/2007/GB003081.

Russel, J. and J.M. Wallace, 2004: Annual carbon dioxide drawdown and the northern annular mode. Glob. Biogeochem. Cy., 18, GB1012, doi:10.1029/2003/GB002044.

Schaefer, K. A.S. Denning, N. Suits et al., 2002: Effect of climate on interannual variability of terrestrial CO2 fluxes. Glob. Biogeochem. Cy., 16(4), 1102, doi:10.1029/2002GB001928.

---

## Author Comment (AC1) · 28 May 2016

Response to referee comments (in red):

Anonymous Referee #3

This paper uses the results from two atmospheric inversion models and long-term surface temperature and NDVI records to compute trends in the CO2 fluxes in the Arctic and Boreal regions (excluding Europe). The authors conclude that the Boreal region has become an increasingly large sink for CO2, with no statistically significant change in the Arctic, even though the seasonal cycle amplitude in CO2 in both regions has increased. The authors argue that this is due to the balance between increased summertime uptake and fall CO2 emissions. The paper is well-written and clear, and suitable for publication in ACP. I recommend that this paper is published after addressing the following comments.

Main Comments:
I would recommend that the authors look at the more recent solar-induced fluorescence (SIF) measurements (e.g., GOME-2, GOSAT, OCO-2) in their analyses. SIF is reported to be more directly related to photosynthesis than greenness indices are, and show some significant differences in the Boreal and Arctic regions (e.g., Joiner et al. 2013). GOME-2 has the longest time series (launched in 2006), and I recognize that this does not cover the main time period of the inversions, but it should be helpful to determine whether NDVI is fully capturing the productivity cycle in the Boreal region.

> We are also excited about the potential of SIF in quantifying carbon fluxes in the high northern latitudes. However, an analysis of SIF and changes in growing season length are outside the scope of this study. This comment was a good reminder to discuss the possible disconnect from NDVI and GPP on a seasonal time scale. This was added: *"Comparisons with recent satellite measurements of solar induced fluorescence show that the seasonality of NDVI may not capture the seasonality in GPP (Walther et al., 2015), but we focus on interannual variability of growing season sums and maximum July values in this study."*
> In this study, NDVI was not used as a model input, so bias in the seasonal cycle will not affect the inversion fluxes calculated.

This analysis does not directly consider the timing of the onset of the growing season, but it is obvious in Figure 3a that even between the two models using the same CO2 concentration data, the phase and duration of the growing season are inconsistent. This raises several questions: Are monthly fluxes temporally fine enough for this analysis (i.e., would the results change if you were to look at, say, bi-weekly fluxes)? Do the two inversions show a similar change in the timing of the onset of the growing season over time? Do they show consistent changes in the length of the growing season?

> This disagreement between the models at the beginning and end of the growing season does raise some interesting questions. The phase of the fluxes is not fixed (held constant) in either model, so there is no obvious explanation for why they would differ, other than the two models are entirely independent of each other. There are other metrics of season start/end such as NDVI and SIF that are better suited to identifying trends in the shoulder seasons if your focus is on productivity (GPP) and not the net $CO_2$ fluxes (which include respiration contributions). The decades long focus of this study limits the spatial and temporal coverage of atmospheric $CO_2$ observations. In the future, including the denser network of atmospheric observing stations, spatially and temporally, should improve the power of atmospheric inversions to quantify start/end of the net $CO_2$ uptake season.

Minor Comments:
Title: I suggest you clarify the title by specifying that the inversions use surface concentrations and that the remote sensing is of NDVI and temperature
> Excellent point. Changed the title.

P2L22:... trigger *a* massive...
> Corrected.

P2L35: Is (1997) referring to a paper?
> Citation error. Fixed it.

P5L2: Be careful to state that GLOBALVIEW-CO2 isn't "data". From the ESRL webpage (http://www.esrl.noaa.gov/gmd/ccgg/globalview/co2/co2_intro.html): "GLOBALVIEWCO2 is derived from atmospheric measurements but contains no actual data."

Agreed.  Deleted 'data'.

P6Para24: Please clarify. I find the first two sentences very confusing.

*Changed this text to: "The atmospheric inversion approach taken in this study is unlikely to reliably separate influences from different longitudinal regions within the latitude bands discussed here.  Our focus on the longest records possible, from sparse atmospheric $CO_2$ observations starting in the 1980s, compromises the spatial resolution of the inversion fluxes.  Rapid atmospheric mixing of a few weeks around latitude bands makes it hard to separate fluxes for example from North America and Eurasia."*

P9L25: In order \*to\* investigate...

Corrected.

P10L6: You show the average growing season NDVI. Would the integrated NDVI over the growing season be better correlated with CO2 uptake?

In this analysis, the "growing season" is defined as April through October, everywhere, so the mean and the integrated NDVI would have the identical correlation.

P10L13: How does the month of the maximum NDVI change over time? Is there a trend?

While the maximum value of NDVI changes, the timing of the maximum does not change.  The focus of this study is really the CO2 fluxes.  Figure 3 shows no indication that the timing of the maximum $CO_2$ uptake has shifted either.

P10L27: How is significance defined here?

We added a paragraph on the statistical methods used in section 2.3.  "*Trends were considered significant if they passed the 90% confidence level (p-values < 0.1)."*

P12L28: ... warm summers may \*be\* driven...

Corrected.

P12L27: Schneising et al. (2014) also came to a similar conclusion.

Added this reference.

P13L22: ... to different \*latitude\* bands...

Corrected.

Figure 3: The two inversions differ in their mean seasonal cycle amplitudes by a factor of two in the Arctic, and they have significantly different onsets of the growing season in the Boreal zone. Can you explain why?

They are 2 entirely independent inversion models and it is not surprising that there are some differences. The modelers involved in this study have not identified a specific cause of the differences, but it likely is related to different prior fluxes and atmospheric transport models.  Also, a simple explanation for some of the model differences is how they split fluxes between boreal and temperate zones.  This makes the fluxes in either zone, and particularly in the arctic zone, with smaller fluxes, somewhat less robust.  A variable amount of leakage of boreal fluxes into the arctic could lead to large changes in the arctic $CO_2$ amplitude.  The inversions are much stronger constraints on interannual variability and trends in the fluxes than on the shape of the $CO_2$ flux seasonality.  Added: "*These differences are not unexpected given the differences in atmospheric transport (including vertical mixing and leakage across latitudes), a priori fluxes, observational network inputs, and model structure between the inversion models.  In this analysis we try to focus on the most robust features were the models do tend to agree on the interannual trends in anomalies from the mean."*

References:

Joiner, J., Guanter, L., Lindstrot, R., Voigt, M., Vasilkov, A. P., Middleton, E. M., Huemmrich, K. F., Yoshida, Y., and Frankenberg, C.: Global monitoring of terrestrial chlorophyll fluorescence from moderate-spectral-resolution near-infrared satellite measurements: methodology, simulations, and application to GOME-2, Atmos. Meas. Tech., 6, 2803-2823, doi:10.5194/amt-6-2803-2013, 2013.

Schneising, O., M. Reuter, M. Buchwitz, J. Heymann, H. Bovensmann, and J. P. Burrows (2014), Terrestrial carbon sink observed from space: variation of growth rates and seasonal cycle amplitudes in

response to interannual surface temperature variability,
Atmos. Chem. Phys., 14(1), 133–141, doi:10.5194/acp-14-133-2014.

Response references:
Walther, S., Voigt, M., Thum, T., Gonsamo, A., Zhang, Y., Koehler, P., Jung, M., Varlagin, A. and Guanter, L.: Satellite chlorophyll fluorescence measurements reveal large-scale decoupling of photosynthesis and greenness dynamics in boreal evergreen forests, Global Change Biology, doi:10.1111/gcb.13200, 2015.

---

## Author Comment (AC2) · 28 May 2016

Response to referee comments (in red):

Anonymous Referee #2

This study takes an important step beyond the well-documented increase in atmospheric CO2 seasonal amplitude at Arctic monitoring sites and asks whether this amplitude increase actually reflects a net gain in CO2 uptake in boreal and Arctic regions. The methodology involves exploring trends in NEE fluxes inferred from 2 different inversion systems over the common period 1986-2006 (and also 1985-2012 for one of the inversions). In general, the study is well presented and documented and I support publication with minor revision. Some of my more important (although still relatively minor) concerns are that the results differ substantially between the two inversions in many aspects, leading to doubts about the robustness of either.

Added: " *These differences are not unexpected given the differences in atmospheric transport and model structure between the inversion models. In this analysis we try to focus on the most robust features were the models do tend to agree on the trends in anomalies from the mean.*"

Also, the Arctic zone>60N is the region with the most unequivocal increase in CO2 amplitude, yet the inversions estimate significant trends in net CO2 uptake mainly in the boreal zone (50-60N), not the Arctic zone. The CO2 amplitude increase at Barrow, AK (71N) in particular has been the subject of much attention, yet it doesn't seem to be associated with an actual increase in CO2 uptake in the surrounding region. A particularly interesting result is that the inversions suggest that increased CO2 respiration and release in fall may largely balance increased CO2 uptake in summer (although they don't agree where the increased fall respiration is occurring). I am curious about the heavy focus on midsummer (July) at the expense of late spring/early summer, when the CO2 cycle (e.g., at Barrow) indicates an earlier onset of photosynthesis. Could this be when some of the net gain in CO2 uptake is occurring?

I think the reviewer is confusing trends in concentration amplitude with flux amplitude. The inversion should be able to separate influence in the spring from the mid-summer.

Re: the 2 time periods chosen: 1986-2006 and 1985-2012. I suggest making the second period 1986-2012, to remove ambiguity about why the results differ between the 2 periods. With the 1985 start year, we don't know whether the changes in the trends are due to the influence of starting in 1985 vs. 1986 or due to more recent changes from 2006-2012. The latter possibility seems more relevant to global change, therefore I suggest eliminating this ambiguity by starting both periods in 1986. Trend calculations of this sort can be sensitive to the starting year, especially when operating on the margins of statistical significance, as is the case here. On a related note, is the legend in Fig 3b (86-12) a typo?

Regarding the start year and periods of trend calculations, we agree with the reviewer's comments about sensitivity to start year. That's why we think it is a more robust estimate to use the longest records possible. It is not our intension to comment on the difference between 86-06 and 85-12 as a measure of processes from 06-12. Rather, the intension to use as much information as possible to examine the longterm trends. Fixed the 86-12 typo. Should be 85-12.

Some specific comments
Abstract, there are a couple of grammatical errors or typos that interfere with smooth reading:
AbL17-18 "Here we examine CO2 fluxes from northern boreal and tundra from 1986 to 2012 ..."
edited

AbL29-31 sentence beginning with "Meanwhile . . .
edited

P2L35 (1997)?
citation fixed

P3L20-26 Please define what exactly is meant by "browning" and "greening," e.g., does this refer to changes in seasonality of NDVI, or does it refer to an annual mean index?
Some studies examine maximum and others the growing season integrated NDVI. This comment was added.

The Introduction in general is quite good and informative, but is marred by the paragraph on L8-17. I have several suggestions for improving it:
P4 L8-17 The emphasis on aboveground vs. belowground in the first sentence seems incongruous because it is

not mentioned earlier as a strength of inversions. Perhaps start this paragraph with a more general statement about the strengths of forest inventories.

The reviewer's comment was valid. We edited the entire paragraph to improve the context with the rest of the introduction.

P4 L13 For clarity, should "several studies" be "several process-based model studies"?

change made

P4 L11-17 Can we believe these results? What are the weaknesses of process-based model studies? (Referring back to earlier statement that, "Each of these methods has its strengths and weaknesses.")

The models need to be validated and atmospheric inversions can help in that effort.

P4L18 ". . .50N, using the atmospheric inversion method."

Moved the second sentence forward to introduce the inversion method at the start.

P4L35 What is "It" ?

RIGC

P5L7 What period?

1985-2006

P5L9 Temporal coverage of what? Years, months, weeks? What is the time resolution?

It varies by station and time period, but at least monthly resolution was the aim.

P5L21 What is LPJ?

The Lund-Potsdam-Jena model is commonly referred to a LPJ in the literature.

P5L30 What are the units of NDVI? Are they mass units, e.g., kg/m2 or flux units, e.g., in kg/m2/s?

Unitless. It's a ratio of light reflectance in different wavelengths. This is described briefly now.

P6L10- Perhaps I am missing something, but I don't see the 2 different analysis methods for trends and significance reported in Table 2 described anywhere in this section. There is only a brief mention of them in the Table 2 caption, which is not very informative.

We added a paragraph in section 2.3, Analysis Approach, that describes each of these statistical methods and cites the sources.

P6L15 In Figure 1 the boreal forest stippling extends well north of 60 degrees. Does this mean that the so-called Arctic zone consists largely of boreal forest? This is somewhat confusing and perhaps should be noted here. Other parts of the text seem to suggest the Arctic zone is mainly tundra, but later p.12 mentions that tundra covers only 25% of the Arctic zone.

Added a comment on this.

Figure 3c,d. Should the Y-axis units be gC/m2/day per year?

Yes. Fixed.

P7L33. Probably should note that P < 0.1 is significant at only 10% level, which is a weak standard. In general p < 0.05 is the standard level required for significance.

Added a comment on this to the new paragraph on statistics in section 2.3.

P9L13. How were these 40-50 and 55-65N bands chosen?
Figure 7 seems to suggest net release and net uptake for 40-55N and 55-75N, respectively. Also, please check P13L10 for consistency.

From comparing July and fall trends in Fig 7b. Changed 55-65 to 55-70N.

P9L25 In order to . . .

Fixed.

P10L27 "We found significantly strong positive correlations between July CO2 flux and April through August temperatures of the same year. . ." The next sentence is confusing because it suggests lower CO2 uptake (more

release) in warm years, in contrast to the quoted sentence – please clarify that "positive correlation" means the July flux is weaker not stronger.

> This is confusion about the sign convention of NEE.  Added: *"It is also important to remember that NEE is negative when there is net $CO_2$ uptake from the atmosphere when interpreting the sign of correlations."*

P12L17-20 "Increased summer CO2 uptake cannot be explained by earlier spring leaf-out, but rather points to changes in mid-summer photosynthetic and respiration fluxes themselves." Where is this sentence supported in the Results?

> We decided to cut this sentence because the point about increased summer uptake was already made. Relating that model prediction to spring leaf out was confusing.

P12L31-33 "This difference could reflect the importance of structural ecosystem changes due to warming on the long time scale increasing photosynthesis (Graven et al., 2013), but on the short time scale, respiration is the dominant control." This seems like a core conundrum of this study (together with the fact that no apparent increase in net CO2 uptake is occurring in the band where the CO2 amplitude is increasing). Both of these points might be worth discussing more.

> Actually, the July $CO_2$ uptake is increasing in the boreal zone, as shown in Fig 5, it's just smaller when expressed as a % increase in the seasonal flux amplitude in Fig 4.   Graven et al. (2013) showed that the summer boreal $CO_2$ uptake must be increasing as well from atmospheric constraints.   Atmospheric transport can cause somewhat of a disconnect between observed amplitude changes and the region of fluxes.  It has been shown that even far northern flask stations are somewhat influenced by more southerly fluxes.
>
> I don't find the different drivers for long-term trends and short-term interannual variability to be contradictory.  Added: "*This difference could reflect the importance of structural ecosystem changes due to warming on the long time scale increasing photosynthesis (Graven et al., 2013), but are also consistent with respiration as the dominant control of NEE on short time scales (Schaefer et al., 2002)."*

Response references:
Graven, H. D., Keeling, R. F., Piper, S. C., Patra, P. K., Stephens, B. B., Wofsy, S. C., Welp, L. R., Sweeney, C., Tans, P. P., Kelley, J. J., Daube, B. C., Kort, E. A., Santoni, G. W. and Bent, J. D.: Enhanced seasonal exchange of $CO_2$ by northern ecosystems since 1960, Science, 341(6150), 1085–1089, doi:10.1126/science.1239207, 2013.

Schaefer, K., Denning, A. S., Suits, N., Kaduk, J., Baker, I., Los, S. and Prihodko, L.: Effect of climate on interannual variability of terrestrial $CO_2$ fluxes, Global Biogeochemical Cycles, 16(4), 49–1–49–12, doi:10.1029/2002GB001928, 2002.

---

## Author Comment (AC3) · 28 May 2016

Response to referee comments (in red):

Anonymous Referee #1

This paper is competently written, and I don't find obvious errors in method, analysis, or results. My main complaint has to do with context and integration of results into previous results.
The authors spend some time reviewing previous, sometimes contradictory studies of the boreal and arctic regions
- conflicting browning/greening NDVI studies
- the 'carbon bomb' vs. the authors' results that don't show a large carbon efflux from permafrost regions
- high northern latitudes have decreasing sink, or even becoming a net source vs. the present study that disagrees with this result

After multiple readings of the paper, I'm not sure how far this work goes towards resolving any of these questions, but I think potential is there to do so. The basic result, that there is increasing CO2 uptake in the boreal region (not in the arctic) while the amplitude of arctic CO2 cycles has increased, seems reasonably well established by the results of their study. What I don't really get is a sense of how these results fit into the literature to confirm or deny other hypotheses as a means to clarify our understanding of this admittedly complex region.

In the introduction the authors say that "The net carbon balance of increased plant growth and increase soil respiration is unclear, but has important consequences for predicting carbon-climate feedbacks." By the end of the paper, I don't get the feeling that the authors make a definitive statement addressing this one way or the other. I believe this study has merit, and that any flaws are not fatal. A more rigorous organization of previous literature and the place of this study within our understanding would be helpful. Also, it seems that perhaps the authors are being too passive and 'nice' here, and are just presenting their results without directly confirming or refuting the work of others. Be bold! In the conclusion, state who among your predecessors you agree with, who you disagree with, and say why. You take the risk of perhaps ruffling a few feathers, but you will ensure response, and that's a very effective way to move science forward. (I'm reminded of a current disagreement between a group that hypothesizes that the Amazon experiences greenup during drought, and the group that believes this isn't the case. The issue has not been resolved, but there have been some very interesting studies that have come out of the dispute.)

We appreciate the reviewer's encouragement to take a stronger position on how our findings relate to the existing body of literature. We find no trends towards the carbon release that is often predicted for this region. Our limited temporal study is however unable to weigh in on whether that carbon release will ever occur in the future. We simply can say it hasn't happened yet. We can say that as a whole, the boreal region is maintaining carbon uptake strength in spite of the often discusses drought effects in Alaska and Canada. It could be that opposite trends in Eurasia are offsetting drought effects in North America. We do not feel comfortable using the inversion fluxes to attribute flux trends longitudinally between NA and EU. There aren't enough CO2 observation stations to constraint this well. We have expanded the conclusions section to draw the reader's attention to areas of conflict with previous literature. Including: "*Furthermore, our atmospheric inversions results show no evidence of an overall trend towards increasing $CO_2$ releases in either the boreal or Arctic zone over the 1985-2012 period. This is an important check for process-based biospheric models which have been challenged to predict the timing of an incipient 'carbon bomb' from the high northern latitudes (Treat and Frolking, 2013). At the moment, the increase in biomass productivity has appeared to be outpacing $CO_2$ losses from warming northern carbon-rich soils. Time will tell whether this trend continues, or whether it will reverse, due to nutrient or water limitations, etc., and become a net carbon source in a few decades as predicted by popular opinion among the community of experts (Abbott et al., 2016).* "

Some specific comments:
- Author is not listed in reference in the 4th paragraph of the introduction.

Fixed.

- The Jena inversion uses LPJ land flux and Mikaloff Fletcher/Takahashi ocean flux. What does the RIGC inversion use?

Added these details to the model description.

How are these surface fluxes similar/different, what might that mean for inversion results? Could these differences be the source of the RIGC peak CO2 uptake being double that of Jena (section 3.1.1)?

These priors are used as a starting point and allowed to change based on the inversion residual minimization. They could contribute to the differences in the inversion results. They also use different atmospheric transport models and entirely different model configurations. It is hard to identify one cause of the differences. The fact that they share many similar trends gives us some confidence that those trends are robust. Inversion models are known to vary widely in their magnitude of the fluxes. For that reason, interannual variability is the focus of our study (Baker et al., 2006).

- Section 3.3: the authors claim that the flux amplitude increase, shown in figures 3cd, is larger in the arctic than in the boreal regions. This is clearly true in the RIGC product, especially with regard to SON efflux. However, I'm not sure I agree that this is true for Jena. To my (subjective) eye, the summer uptake and fall/winter efflux amplitude increase is larger for both Jena products in the boreal region than in the arctic.

It's roughly the same absolute increase in the flux amplitude, but that's on top of very different mean seasonal cycles. It's the percentage increase that larger in the Arctic region.

- I'm a bit confused about the results shown in sections 3.5 and 3.6, Figure 11. Figure 3 clearly displays a strong amplification of July CO2 uptake, and Figure 8 shows a clear upward trend in JJA temperatures over the period of study. But Figure 11 (and references to studies in the text) correlate cooler summertime temperatures with increased uptake. What am I missing here? These seem contradictory. Is the moisture component the more important than the temperature?

The difference is that the records were detrended before the correlation analysis in Fig 11. There may be different drivers of long-term trends and short-term interannual variability.
*"In this analysis, all data sets were de-trended using a stiff spline to remove long-term trends, thus emphasizing processes controlling interannual variability (IAV)."*

- Section 3.5: Russell and Wallace (2004) and Schaefer et al. (2002) looked at carbon flux in relation to modes of climate variability such as the annular modes. Hurrell et al. (2001) discussed trends in the NAO itself. Would studies such as these help provide context here, or are they unrelated?

The annual modes are related in that they correspond to temperature and precipitation anomalies, but I don't think it's necessary to include them in the discussion. The analysis of temperature controls on NEE and NDVI is fundamental regardless of whether the temperature anomaly is caused by an annual mode or not. We did add a statement related to the Russell and Wallace findings…
*"The RIGC inversion shows significant correlation between warm winters and increased CO$_2$ uptake the following growing season (negative correlation), consistent with Russell and Wallace (2004), but this relation did not appear in the Jena correlation."*

- Is the last paragraph of section 3.6 necessary?

It seems important for completeness, and it's interesting that the same patterns don't hold in the northern region.

- Figure S1: RIGC BA+BNA fossil fuel (ORNL/EDGAR) is about half the Jena anthropogenic flux for the same region (also EDGAR, but apparently different version.
Intuitively, I would expect that Jena uptake would have to be larger than RIGC to resolve observed CO2 concentration with these anthropogenic fluxes. Why isn't this the case?

*"The RIGC and Jena inversions use different fossil-fuel emissions datasets to isolate the net land surface fluxes related to biology. Comparing fossil-fuel emissions for the EU and BA+BNA Arctic and Boreal zones used in each inversion (SI Fig. 1) shows that while the mean emissions were lower in the RIGC inversion,*

*the IAV and trends in absolute fluxes were similar in each inversion. Differences in the fossil emissions are therefore unlikely to contribute significantly to trends in the biological land fluxes of the BA+BNA Arctic and Boreal zones."*

Remember that the long-term means have been removed in this analysis because of offsets like this among models.

- Patra et al. (2008) and Parazoo et al. (2008) discuss model resolution in relation to simulations of CO2. I wonder if advection of the effect of large surface CO2 flux into boreal/arctic regions is a partial (or dominant?) cause of the increasing amplitude of high-latitude CO2 concentrations? Or is Graven et al. (2013) the last word? What role might model resolution play? Are these issues not germane to this manuscript?

There is a body of literature suggesting that many transport models under estimate the vertical mixing, which would directly affect these inversion predictions, in particular when the measurement sites are located close to the intense source regions, e.g., the land biosphere or industrial centers. However, the fact that 2 different transport models produce similar $CO_2$ flux trends is reassuring. This consistency between the two inversions is obtained because the measurement sites used in both inversions are remotely located and designed to sampling marine air. Patra et al. (2008) have shown that the so called "site representation error" is high for the coastal or continental sites.

- Figure 11: There are significant correlations out to two years for RIGC and 4 years for Jena that are not discussed in the text. What might these long time-lag correlations mean?

Added this paragraph: *"Our analysis found significant correlations out to 2 to 4 years prior, suggesting that temperature anomalies could have an impact on NEE after several years delay. While there have been studies suggesting that multi-year lags between climate and $CO_2$ fluxes are important (e.g. Bond-Lamberty et al., 2012), these correlations were not consistent between the inversion models, preventing us for speculating as to the cause. "*

References
Hurrell, J.W., Y. Kushnir, M. Visbeck 2001: The north atlantic oscillation. Science, Vol. 291, No. 5504 (Jan 26 2001), p603-605.
Parazoo, N.C., A.S. Denning, S.R. Kawa et al., 2008: Mechanisms for synoptic variations of atmospheric CO2 in North America, South America, and Europ. Atmos. Chem. Phys., 8, 7239-7254.
Patra, P.K., R.M. Law, W. Peters, et al., 2008: TransCom model simulation of hourly atmospheric CO2: Analysis of synoptic-scale variation for the period 2002-2003. Glob. Biogeochem. Cy., 22, GB4013, doi:10.1029/2007/GB003081.
Russel, J. and J.M. Wallace, 2004: Annual carbon dioxide drawdown and the northern annular mode. Glob. Biogeochem. Cy., 18, GB1012, doi:10.1029/2003/GB002044.
Schaefer, K. A.S. Denning, N. Suits et al., 2002: Effect of climate on interannual variability of terrestrial CO2 fluxes. Glob. Biogeochem. Cy., 16(4), 1102, doi:10.1029/2002GB001928.
* * *
Response references:
Baker, D. F., Law, R. M., Gurney, K. R., Rayner, P., Peylin, P., Denning, A. S., Bousquet, P., Bruhwiler, L., Chen, Y.-H., Ciais, P., Fung, I. Y., Heimann, M., John, J., Maki, T., Maksyutov, S., Masarie, K., Prather, M., Pak, B., Taguchi, S. and Zhu, Z.: TransCom 3 inversion intercomparison: Impact of transport model errors on the interannual variability of regional $CO_2$ fluxes, 1988–2003, Global Biogeochemical Cycles, 20(1), GB1002, doi:10.1029/2004GB002439, 2006.

Bond-Lamberty, B., Bunn, A. G. and Thomson, A. M.: Multi-year lags between forest browning and soil respiration at high northern latitudes, edited by G. Bohrer, PLoS ONE, 7(11), e50441, doi:10.1371/journal.pone.0050441.t002, 2012.

Patra, P. K., Law, R. M., Peters, W., Rodenbeck, C., Takigawa, M., Aulagnier, C., Baker, I., Bergmann, D. J., Bousquet, P., Brandt, J., Bruhwiler, L., Cameron-Smith, P. J., Christensen, J. H., Delage, F., Denning, A. S., Fan, S., Geels, C., Houweling, S., Imasu, R., Karstens, U., Kawa, S. R., Kleist, J., Krol, M. C., Lin, S. J., Lokupitiya, R., Maki, T., Maksyutov, S., Niwa, Y., Onishi, R., Parazoo, N., Pieterse, G., Rivier, L., Satoh, M., Serrar, S., Taguchi, S., Vautard, R., Vermeulen, A. T. and Zhu, Z.: TransCom model simulations of hourly atmospheric

$CO_2$: Analysis of synoptic-scale variations for the period 2002-2003, Global Biogeochemical Cycles, 22(4), n/a–n/a, doi:10.1029/2007GB003081, 2008.

Russell, J. L. and Wallace, J. M.: Annual carbon dioxide drawdown and the Northern Annular Mode, Global Biogeochemical Cycles, 18(1), n/a–n/a, doi:10.1029/2003GB002044, 2004.